# HIV efficiently infects T cells from the endometrium and remodels them to promote systemic viral spread

Tongcui Ma[1,2], Xiaoyu Luo[1], Ashley F George[1,2], Gourab Mukherjee[3], Nandini Sen[4], Trimble L Spitzer[5], Linda C Giudice[6], Warner C Greene[1,7], Nadia R Roan[1,2]*

[1]Gladstone Institute of Virology and Immunology, San Francisco, United States; [2]Department of Urology, University of California, San Francisco, San Francisco, United States; [3]Department of Data Sciences and Operations, University of Southern California, Los Angeles, United States; [4]Departments of Pediatrics and Microbiology and Immunology, Stanford School of Medicine, Stanford, United States; [5]Lt Col, United States AF; Women's Health Clinic, Naval Medical Center, Portsmouth, United States; [6]Center for Reproductive Sciences, Department of Obstetrics, Gynecology and Reproductive Sciences, University of California, San Francisco, San Francisco, United States; [7]Department of Medicine, University of California, San Francisco, San Francisco, United States

*For correspondence:
nadia.roan@ucsf.edu

Competing interests: The authors declare that no competing interests exist.

**Abstract** The female reproductive tract (FRT) is the most common site of infection during HIV transmission to women, but viral remodeling complicates characterization of cells targeted for infection. Here, we report extensive phenotypic analyses of HIV-infected endometrial cells by CyTOF, and use a 'nearest neighbor' bioinformatics approach to trace cells to their original pre-infection phenotypes. Like in blood, HIV preferentially targets memory CD4+ T cells in the endometrium, but these cells exhibit unique phenotypes and sustain much higher levels of infection. Genital cell remodeling by HIV includes downregulating TCR complex components and modulating chemokine receptor expression to promote dissemination of infected cells to lymphoid follicles. HIV also upregulates the anti-apoptotic protein BIRC5, which when blocked promotes death of infected endometrial cells. These results suggest that HIV remodels genital T cells to prolong viability and promote viral dissemination and that interfering with these processes might reduce the likelihood of systemic viral spread.

## Introduction

Sexual contact remains the most common mode of HIV transmission, and more than half of new worldwide infections occur in women. Most new cases arise in sub-Saharan Africa, where in 2018 four out of every five newly infected individuals aged 15–19 were female (*UNAIDS, 2019*). The typical initial site of HIV acquisition in women is the female reproductive tract (FRT). Although HIV is initially deposited into the lower FRT (vagina and ectocervix), peristalsis propels vaginal lumen contents such as HIV into the upper FRT (endocervix, endometrium, and fallopian tubes) (*Barnhart et al., 2001*; *Egli and Newton, 1961*; *Hartman, 1957*; *Kunz et al., 1997*; *Leyendecker et al., 1996*; *Stieh et al., 2014*). There, local exposure to progesterone may further increase the likelihood of infection by diminishing epithelial barrier function (*Neidleman et al., 2017*). Consistent with both the upper and lower FRT being potential sites of transmission, SIV

vaginal infection studies demonstrate that transmission can initiate throughout the entire FRT (*Stieh et al., 2014*).

Once infected, susceptible cells from the FRT become the founder population that eventually leads to systemic spread of the virus. A limited number of studies have queried the features of these initial HIV targets. Implementation of the FACS-based virion fusion assay – which identifies cells that HIV has entered (*Cavrois et al., 2002*) – on endocervical specimens revealed preferential entry of HIV into cells with higher levels of the CCR5 co-receptor and the activation/T resident memory (Trm) marker CD69 (*Joag et al., 2016*). A more recent study examining HIV fusion to cells isolated from endometrial and endocervical biopsies also found higher fusion to CD4+ T cells expressing CD69 (*Cavrois et al., 2019*). Because fusion does not always lead to productive infection due to the activity of various post-entry viral restriction factors (*Stevenson et al., 1990*; *Zack et al., 1990*), it is important to complement HIV fusion studies with assays that identify productively-infected cells.

To this end, FACS has been used to characterize expression levels of several cell surface receptors on FRT-derived cells productively-infected in vitro with HIV (*Rodriguez-Garcia et al., 2014*; *Swaims-Kohlmeier et al., 2016*). These studies revealed interesting properties of HIV-infected FRT cells, but one limitation is their inability to determine whether such features are selected by HIV *before* infection, or changed by HIV via remodeling *after* infection. Indeed, HIV and other viruses markedly remodel cells by up- or down-regulating a variety of cell-surface receptors (*Cavrois et al., 2017*; *Sen et al., 2015*), the classic example being the well-characterized decrease in surface expression of CD4 (*Garcia and Miller, 1991*; *Vincent et al., 1993*). To distinguish between preferential infection versus remodeling, we recently implemented the bioinformatics approach SLIDE (*Sen et al., 2015*) on HIV-infected tonsillar CD4+ T cells phenotyped with a 38-parameter CyTOF panel (*Cavrois et al., 2017*). CyTOF, also known as mass cytometry, is a hybrid between mass spectrometry and flow cytometry that uses antibodies conjugated to metal lanthanides to quantify the expression levels of protein antigens on or within cells (*Bendall et al., 2011*). Because spectral overlap is not a limitation, CyTOF panels can be quite large allowing for deep phenotyping of individual cells. By matching the high-dimensional CyTOF profile of each HIV-infected cell to an 'atlas' of uninfected CD4+ T cells from the same donor, we are able to predict the phenotype of the original cell preferentially targeted for infection (*Cavrois et al., 2017*). Predictions made by this approach, which we term "Predicted Precursor as determined by SLIDE (PP-SLIDE)", were experimentally confirmed through sorting experiments (*Cavrois et al., 2017*). In the current study, we employ a new and validated 38-parameter CyTOF panel tailored for genital T cells and implement PP-SLIDE to characterize the initial cells infected by HIV and the changes that take place in these cells. Our analysis – the first to analyze cells from the FRT by CyTOF – reveal that HIV efficiently infects these cells and remodels them in ways that favor prolonged cell survival and dissemination of the virus to lymphoid follicles within lymph nodes.

## Results

### Endometrial cells are highly susceptible to infection by CCR5-tropic HIV

While CD4+ T cells from unstimulated PBMCs are poorly permissive to productive infection by HIV, a fraction of these cells within tonsils are efficiently infected in the absence of ex vivo stimulation (*Glushakova et al., 1997*). To determine whether genital T cells are similarly permissive in the absence of stimulation, we exposed single-cell suspensions of cells isolated from endometrial biopsies to the HIV-F4.HSA reporter virus. This replication-competent, Nef-sufficient virus harbors the 109FPB4 transmitted/founder (T/F) CCR5-tropic Env, and encodes an LTR-driven heat-stable antigen (HSA) cell-surface protein enabling identification of productively-infected cells by either FACS or CyTOF (*Cavrois et al., 2017*). To boost infection rates, the semen-derived viral enhancer SEM86 (*Roan et al., 2014*) was added to all specimens (see Materials and Methods). Three days after exposure to the reporter virus or media alone as a negative control, cells were analyzed by FACS. CD4+ T cells were identified as live, singlet CD3+CD8- cells, to include cells that have downregulated cell-surface CD4 due to the activity of HIV accessory genes (*Garcia and Miller, 1991*; *Vincent et al., 1993*). A distinct population of CD3+CD8-CD4$^{low}$HSA+ cells was observed in the HIV-exposed but not mock-treated culture, suggesting that productive infection of endometrial T cells (hereafter referred to as ETs) does not require ex vivo stimulation (*Figure 1A*). To compare infection levels in

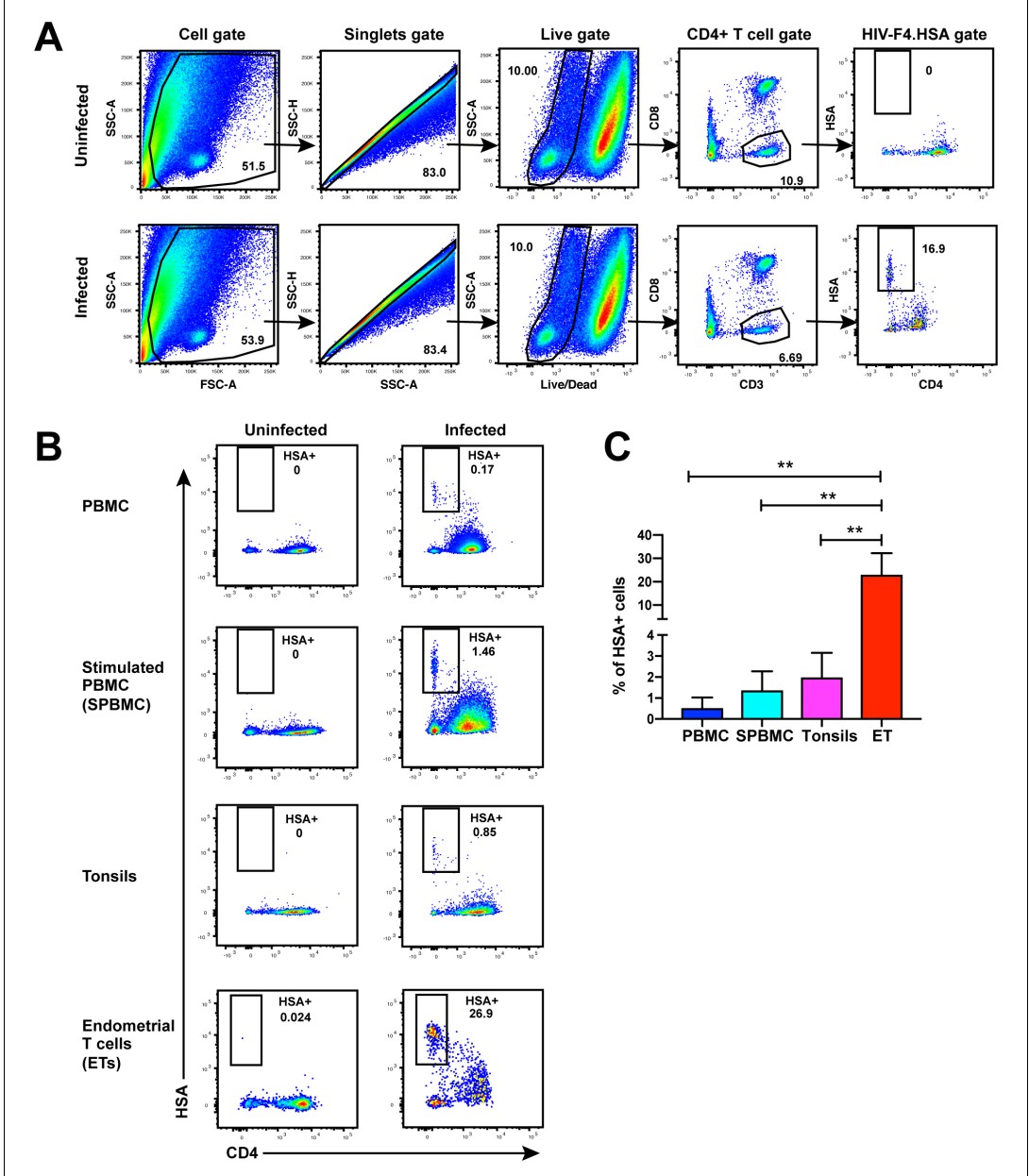

**Figure 1.** Unstimulated endometrial T cells are highly susceptible to infection by CCR5-tropic HIV. (**A**) FACS plots demonstrating a distinct population of productively-infected endometrial T cells (ETs) following exposure to the HIV reporter strain F4.HSA. Unstimulated ETs were mock-treated or exposed to F4.HSA for 3 days and then analyzed by FACS. Shown is the gating strategy leading to identification of HIV-infected ETs, representative of 4 independent donors. (**B**) Representative FACS plots of uninfected and infected cultures of PBMCs (unstimulated or PHA-stimulated), unstimulated tonsillar human lymphoid aggregate (HLAC) cultures, or unstimulated ETs. Cells were mock-treated or inoculated with F4.HSA, and monitored 3 days later by FACS for levels of productive infection. *Left*: Uninfected controls. *Right*: Infected cultures. Results are gated on live, singlet CD3+CD8- cells and are representative of 4 independent donors. (**C**) ETs are significantly more susceptible than PBMCs and tonsils to infection by F4.HSA. Results of FACS analysis from 4 independent donors gated on live, singlet CD3+CD8- cells. \*\*p<0.01 as assessed using the Student's unpaired t test.

The online version of this article includes the following figure supplement(s) for figure 1:

**Figure supplement 1.** Transduction-enhancing fibrils promote HIV infection of endometrial T cells.

ETs vs. T cells from other sites, we exposed equivalent numbers of cells from unstimulated and stimulated PBMCs and from tonsils (through generation of human lymphoid aggregate cultures, or HLACs [*Cavrois et al., 2017*]) to the same dose of F4.HSA in the presence of SEM86. As expected, infection rates were higher in tonsils and stimulated PBMCs than in unstimulated PBMCs. Remarkably, infection rates were more than an order of magnitude higher in ETs than in any of the other tested cells (*Figure 1B and C*).

## HIV-susceptible cells from ETs are mostly effector memory T cells with phenotypic features distinct from those in blood

Having demonstrated by FACS that ETs are highly susceptible to HIV infection, we next characterized the phenotypic features of these cells using CyTOF. To compare directly the highly-permissive ETs to the poorly-permissive blood T cells from the same individuals, we recruited four uninfected women to donate both PBMCs and endometrial biopsies during the same study visit (*Supplementary file 1*). These cells were immediately processed for infection with F4.HSA followed by CyTOF analysis, using a specially designed and validated 38-parameter panel (*Supplementary file 2*, Materials and Methods). A gating strategy identifying live, singlet CD3 +CD8- infected T cells was implemented (*Figure 2—figure supplement 1*). As expected from the FACS data, infection rates were markedly higher in ETs (range 5.1–37%) than in PBMCs (range 0.09–0.46%) (*Figure 2A*).

To compare the uninfected and infected cells at a global level, we implemented t-distributed stochastic neighbor embedding (t-SNE) (*van der Maaten and Hinton, 2008*). This visualization method places phenotypically similar cells close together and can be used to identify local 'islands' of related cells. For both PBMCs and ETs, T cells from the uninfected culture (in aqua) segregated away from infected T cells from the infected culture (in magenta) (*Figure 2B*), suggesting HIV-induced remodeling. When the PBMC and endometrial samples were visualized within the same t-SNE space, it was further apparent that the infected cells from PBMCs and ETs were phenotypically distinct (*Figure 2C* and *Figure 2—figure supplement 2A*). To quantitatively compare the extent of remodeling in the two types of samples, we implemented SLIDE (*Sen et al., 2015*) to assess the distances between infected and uninfected cells in each sample (see Materials and Methods). This analysis revealed that significant remodeling occurred in all infected samples, with no statistically significant difference in the extent of remodeling between PBMCs and ETs (*Figure 2—figure supplement 3*, *Supplementary file 3*). In contrast to the infected cells, bystander (HSA-) Tm cells from infected cultures exhibited significantly lower SLIDE scores, and by t-SNE resembled Tm cells from the uninfected culture (*Figure 2—figure supplement 4A,B*). Furthermore, transfer of supernatants from infected cultures to uninfected cultures in the presence of antiretroviral drugs to prevent infection did not lead to marked remodeling of the Tm cells (*Figure 2—figure supplement 4C,D*). These results suggest that productive infection, and not just an inflammatory milieu created by HIV replication, is required for the extensive remodeling that we observed.

To characterize the HIV-susceptible cells without the confounding effects of HIV-induced remodeling, we identified the Predicted Precursor ('PRE') cell for every HIV-infected cell using PP-SLIDE (*Cavrois et al., 2017*; *Sen et al., 2015*). This method, schematized in *Figure 2D*, assumes that a subset of the infected cell's original identity markers is retained in a manner that can be captured by high-parameter phenotypic analyses such as that offered by CyTOF. By using a k-nearest neighbor approach to identify, for every infected cell, the phenotypically most similar T cell in the uninfected culture, we obtain a set of PRE cells that harbor the predicted phenotypes of the original cells targeted for HIV infection (*Figure 2D*). Comparing the PRE cells to the entire population of uninfected T cells enables us to identify the features of cells that are preferentially targeted for infection by HIV.

After converting all the infected cells to their corresponding PRE cells, we assessed what T cell subsets they belonged to: memory CD4+ T cells (Tm), naïve CD4+ T cells (Tn), memory CD8+ T cells, or naïve CD8+ T cells. The PRE cells from both PBMCs and ETs were almost exclusively Tm cells (*Figure 2E* and *Figure 2—figure supplement 2B*). Among the Tm cells from both blood and the endometrium, effector memory T cells (Tem) were preferentially infected while central memory T cells (Tcm) were largely spared (*Figure 2F*). To confirm these PP-SLIDE predictions experimentally, we sorted Tm cells based on surface expression of the Tcm marker CCR7 and infected them with F4. HSA. Endometrial biopsies yielded too few cells for this assay, but we successfully sorted Tm cells with different levels of CCR7 expression from PBMCs (*Figure 2—figure supplement 5A*). Infection

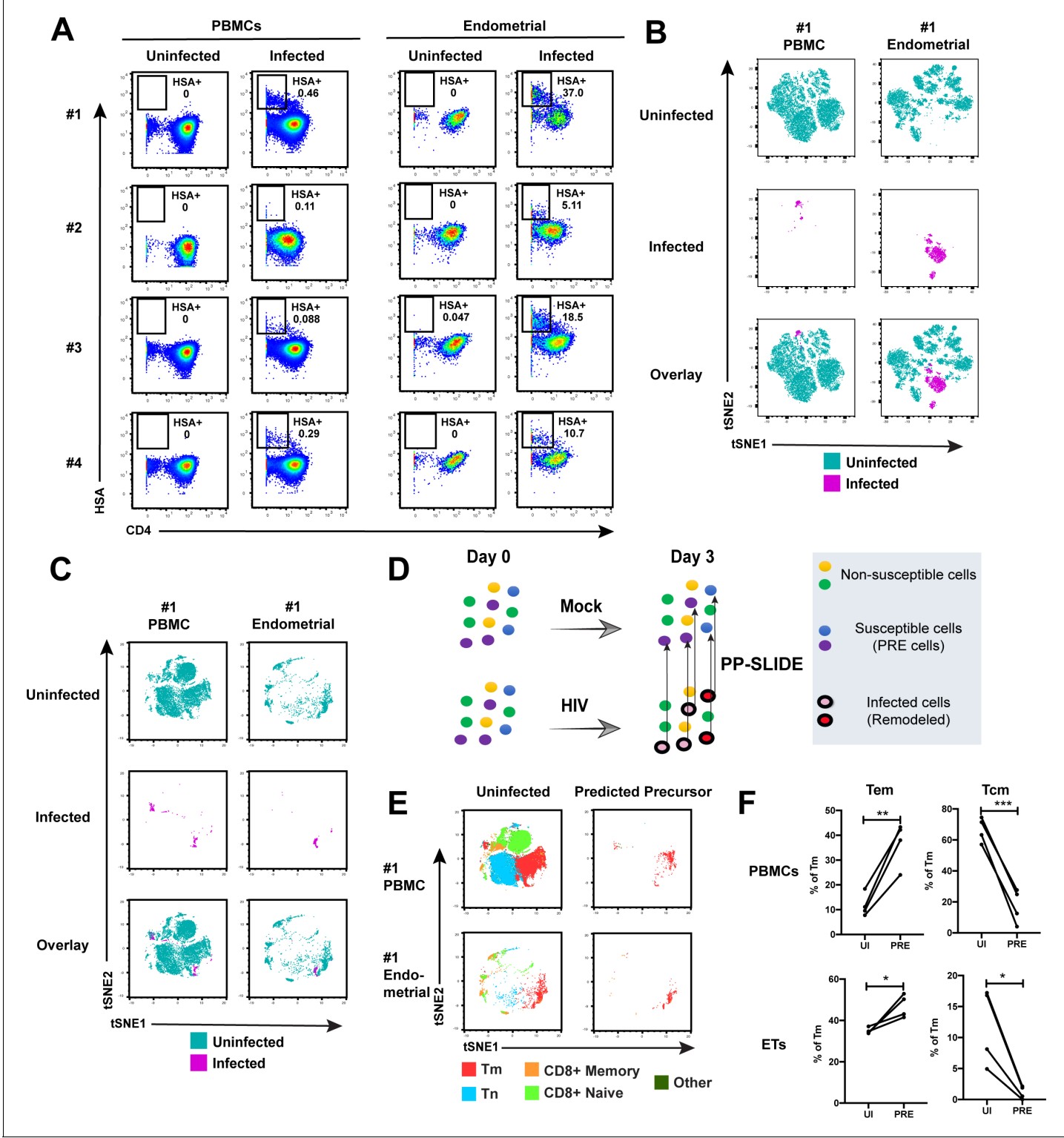

**Figure 2.** Comparison of HIV-susceptible cells in unstimulated PBMCs and endometrium. (A) Paired unstimulated PBMCs and endometrial cells from 4 donors were mock-treated or inoculated with F4.HSA and monitored 3 days later for levels of productive infection. Results are gated on live, singlet CD3+CD8- cells. Numbers correspond to the percentage of infected cells in each sample. (B) t-SNE plots of uninfected and infected cells from Donor #1 shown in panel A, where the PBMC and endometrial specimens were run as separate t-SNEs. Infected cells reside in unique regions of the t-SNE as a result of viral-induced remodeling. Data are representative of 4 independent donors. (C) t-SNE plots of uninfected and infected cells from Donor #1 shown in panel A, where the PBMC and endometrial specimens were run within the same t-SNE. Although the same number of HIV-infected ETs are

*Figure 2 continued on next page*

*Figure 2 continued*

shown here as in panel **B**, these cells occupy less t-SNE space since much of the space is now taken up by cells from the PBMC sample. Corresponding data for the remaining 3 donors in panel **A** are presented in *Figure 2—figure supplement 2*. (**D**) Schematic defining Predicted Precursor ('PRE') cells. A population of T cells is either mock-treated (*top*) or exposed to HIV (*bottom*). These cells include ones that are not susceptible to infection (yellow and green) and those that are (blue and purple). After 3 days, infected cells (surrounded by a black ring) are identified as those expressing the HIV reporter HSA. These cells, however, are remodeled as represented by the conversion from their original blue and purple colors to the red and pink colors, respectively. By using PP-SLIDE to identify for every infected cell the phenotypically most similar cell in the uninfected culture (upward arrows), we identify the PRE cells which harbor the predicted phenotypes of the original cells targeted for HIV infection. These cells do not have the confounder of HIV-induced phenotypic changes, and can be compared to non-susceptible cells in the uninfected culture to identify unique features associated with cells targeted for infection. (**E**) t-SNE plots of uninfected and PRE cells demonstrate that the dominant population of T cells targeted for infection in both blood and endometrium are memory CD4+ T (Tm) cells. Tm cells (CD4+CD45RO+CD45RA-), naïve CD4+ T cells (CD4+CD45RO-CD45RA+, abbreviated Tn cells), memory CD8+ T cells (CD8+CD45RO+CD45RA-), and naïve CD8+ T cells (CD8+CD45RO-CD45RA+) were identified by manual gating and colored as indicated. Corresponding data for the remaining 3 donors in panel **A** are presented in *Figure 2—figure supplement 2*. (**F**) Comparison of the proportions of Tem (CD4+CD45RO+CD45RA-CCR7-CD62L-) and Tcm (CD4+CD45RO+CD45RA-CCR7+CD62L+) among uninfected Tm and PRE cells in PBMCs and ETs reveals preferential infection of Tem in both compartments. *p<0.05, **p<0.01, ***p<0.001 as assessed using the Student's paired t test.

The online version of this article includes the following figure supplement(s) for figure 2:

**Figure supplement 1.** CyTOF gating strategy to identify uninfected and HIV-infected T cells.
**Figure supplement 2.** Comparison of HIV-susceptible cells in unstimulated PBMCs and endometrium.
**Figure supplement 3.** Quantitation of viral-induced remodeling of T cells from PBMCs and ETs.
**Figure supplement 4.** Comparison of uninfected and bystander cells from PBMCs and ETs.
**Figure supplement 5.** Validation of susceptibility of Tm subsets to HIV infection.
**Figure supplement 6.** Antigens differentially expressed between PRE cells from PBMCs vs endometrium.
**Figure supplement 7.** HIV-susceptible Tm cells from endometrium are less activated than HIV-susceptible Tm cells from stimulated PBMCs.
**Figure supplement 8.** Validation of CyTOF antibodies through comparison of antigen expression on immune subsets.

rates inversely correlated with CCR7 expression levels (*Figure 2—figure supplement 5B*), validating the PP-SLIDE prediction. Direct comparison of the frequencies of Tn, Tem, and Tcm subsets in the specimens analyzed by CyTOF revealed Tn and Tcm frequencies to be higher in blood, and Tem frequencies to be higher in the endometrium (*Figure 2—figure supplement 5C*). These data further support the notion that ETs may be more susceptible to F4.HSA infection than their blood counterparts because of the higher frequencies of the highly-susceptible Tem subset.

Although the HIV-susceptible cells in blood and the endometrium both belonged primarily to the Tem subset, they differed markedly in their phenotypes (*Figure 2—figure supplement 6*). Compared to their blood counterparts, PRE cells from the endometrium expressed higher levels of the CCR5 co-receptor and markers of T cell activation such as CD38 and PD1. Markers of Th1 and Th2 cells (Tbet and CRTH2, respectively) were elevated as well, suggesting that HIV-susceptible cells among ETs may be more polarized than their blood counterparts. PRE cells also expressed elevated levels of Blimp-1 and BIRC5, which may facilitate the ability of these cells to support productive infection by promoting HIV transcription and enhanced survival of the infected cells, respectively (*Kaczmarek Michaels et al., 2015*; *Kuo et al., 2018*).

## Clustering reveals a diverse array of HIV-susceptible Tm cells in endometrium but not blood

We next implemented a complementary approach that incorporates all phenotyping parameters instead of manual gating in order to define cellular subsets. Using FlowSOM (*Van Gassen et al., 2015*) to cluster the paired PBMC and ET datasets, we identified twenty clusters among the uninfected Tm cells from the two compartments (*Figure 3A* and *Figure 3—figure supplement 1A*). The PBMC PRE cells resided only within a limited subset of the PBMC clusters (*Figure 3A* and *Figure 3—figure supplement 1A*, left), while endometrial PRE cells were present throughout the region of the t-SNE harboring the endometrial clusters (*Figure 3A* and *Figure 3—figure supplement 1A*, right). Accordingly, the distribution of the 20 clusters was similar between the endometrial total Tm and endometrial PRE cells, but markedly different between the blood total Tm and blood PRE cells (*Figure 3B* and *Figure 3—figure supplement 1B*). Quantification of these results using a chi-

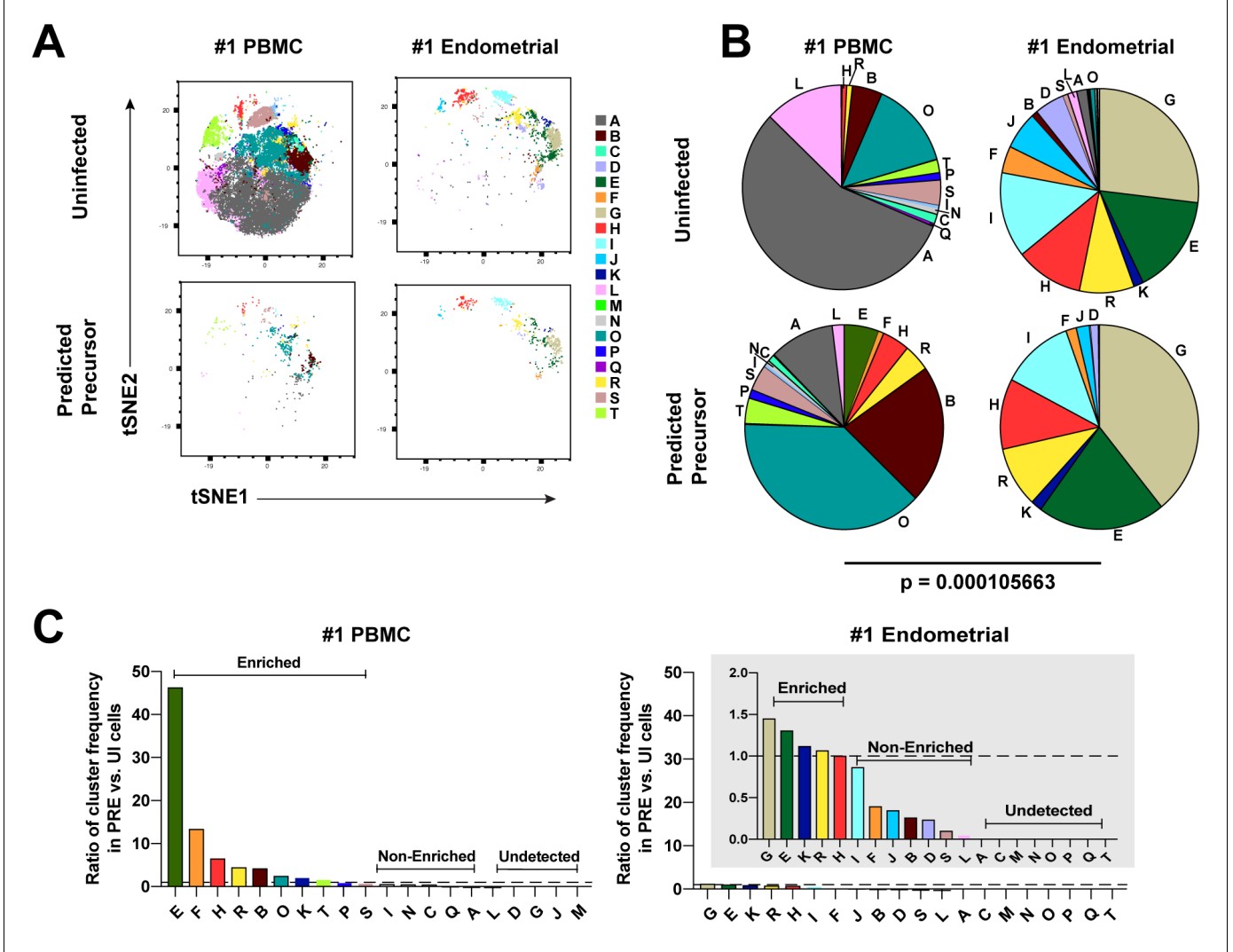

**Figure 3.** Most subsets of Tm in endometrium but not blood are susceptible to HIV infection. (**A**) Tm and PRE cells from the blood and endometrium of Donor 1 were analyzed by FlowSOM, which identified 20 clusters of cells (A – T), each depicted with a different color. Note the pattern of uninfected Tm is different from that of the PRE cells for PBMCs, but similar for the endometrium. (**B**) Pie charts showing proportion of each cluster in the uninfected and PRE cells from PBMCs and ETs. Small clusters with frequencies less than 1% are not depicted. The patterns of clusters in Tm and PRE cells from ETs match more closely than those from PBMCs. Chi-square goodness of fit tests for checking similarity in the distribution of clusters among the Tm and the PRE populations was calculated and the difference in this chi-square dissimilarity metric between the PBMC versus ET datasets is reported as a P-value using procedures detailed in Materials and Methods. (**C**) PRE cells from PBMCs are more enriched for specific clusters than PRE cells from endometrium. To determine whether PRE cells preferentially resided within any of the clusters, the frequency of each PRE cluster was divided by the frequency of the equivalent cluster among uninfected Tm cells (see Materials and Methods). Ratios > 1 indicate clusters that are enriched in PRE cells relative to uninfected Tm cells. Note the extent of enrichment, as reflected by the ratio values, was more pronounced in PBMCs (left) relative to the endometrium (right). The grey inset is a zoomed-in view of the endometrium data. Corresponding data for the remaining 3 donors are presented in *Figure 3—figure supplement 1*.

The online version of this article includes the following figure supplement(s) for figure 3:

**Figure supplement 1.** Most subsets of Tm in endometrium are susceptible to HIV infection.

squared test confirmed that the preferential selection of subsets for infection in PBMCs is significantly higher than that in ETs (*Figure 3B* and *Figure 3—figure supplement 1B*).

We then assessed to what extent each of the identified clusters was enriched among the PRE cells relative to Tm. A cluster was considered to be enriched when the ratio of its frequency in PRE versus Tm cells was >1, with a higher ratio indicating a greater enrichment (see Materials and Methods for details). Analysis of the data from Donor 1 revealed that while PBMCs had ten enriched clusters, ETs

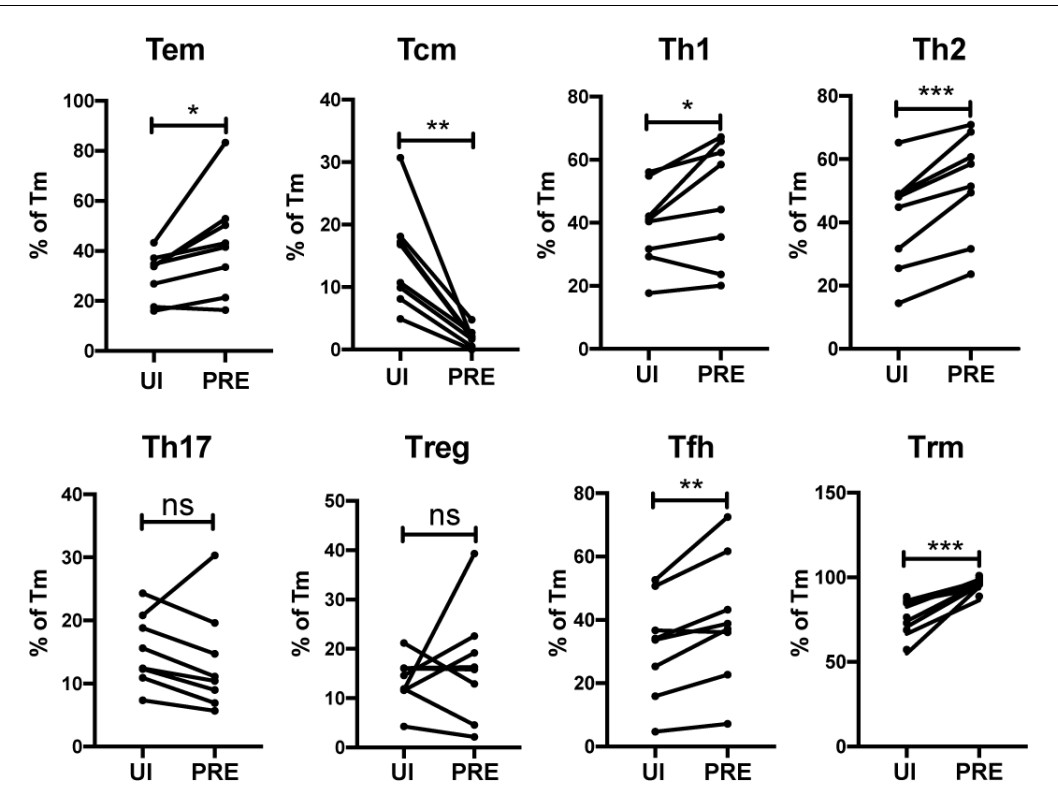

**Figure 4.** HIV preferentially infects endometrial Tem and Trm with phenotypic features of Th1, Th2, and Tfh cells. The proportions of ETs that were Tem (CCR7-CD62L-), Tcm (CCR7+CD62L+), Th1 (Tbet+), Th2 (CRTh2+), Th17 (RORγt+), Tregs (CD127-CD25+), Tfh (PD1+CXCR5+), and Trm (CD69+) were compared among uninfected Tm cells (UI) and PRE cells. *p<0.05, **p<0.01, ***p<0.001 as assessed using the Student's paired t test. n.s.: not significant.

The online version of this article includes the following figure supplement(s) for figure 4:

**Figure supplement 1.** Memory CD4+ T cells from the endometrium of 4 additional donors are highly susceptible to productive infection by HIV.

**Figure supplement 2.** Frequencies of differentiated subsets expressing the Trm marker CD69.

only had five. More importantly, the PBMC clusters were enriched up to more than 40-fold, while the ET clusters were all enriched less than 1.5-fold (*Figure 3C*). Similar results were seen in the 3 additional donors (*Figure 3—figure supplement 1C*). All together, these data demonstrate that HIV only infects a small subset of Tm cells in blood, but conversely a diverse array of Tm cells in the endometrium.

## T cell activation does not fully account for the high susceptibility of endometrial T cells to HIV infection

We next addressed whether the high permissivity of ET Tm cells was associated with a heightened state of T cell activation. Compared to their blood counterparts, ET Tm cells expressed higher levels of multiple activation markers (*Figure 2—figure supplement 7A*) and more frequently expressed the CCR5 co-receptor (*Figure 2—figure supplement 7B*). To probe further whether the high susceptibility of ETs to infection was due to an increased state of T cell activation, we compared the features of HIV-susceptible cells from ETs to those of ex vivo stimulated T cells. ETs, unstimulated PBMCs, or PHA-stimulated PBMCs as a source of activated T cells, were inoculated with F4.HSA and processed for CyTOF analysis after 3 days. As expected, the infection rate was lowest in the unstimulated PBMCs. Interestingly, although the infection rate in stimulated PBMCs was an order of

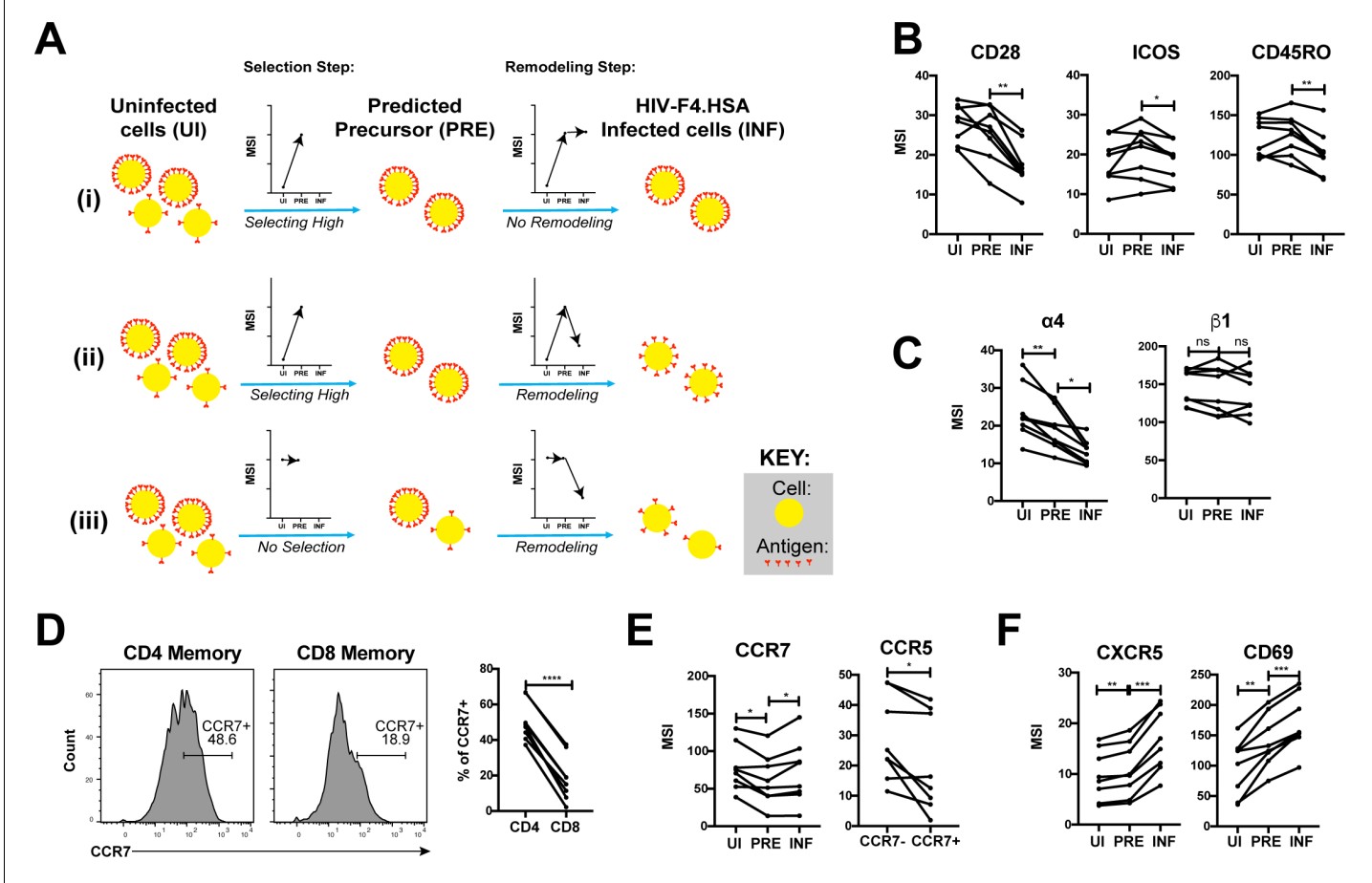

**Figure 5.** HIV remodels cells to impair TCR signaling and promote migration of infected cells to lymph node follicles. (A) Schematic of the use of PP-SLIDE to distinguish HIV-induced up- or down-regulation of an antigen from preferential infection of cells expressing higher or lower levels of the antigen. In this schematic, circles (yellow) correspond to individual cells expressing different levels of a hypothetical antigen (red). The y-axis of the graph reflects the MSI of the antigen. In scenario (i), HIV preferentially infects cells with high levels of the antigen (as reflected by the MSI being higher on PRE cells than uninfected (UI) cells) and doesn't modulate antigen levels after infection (as reflected by the MSI being the same on PRE cells and infected (INF) cells). In scenario (ii), HIV preferentially infects cells with high levels of the antigen (as in scenario (i)), but downregulates its expression in infected cells (as reflected by the MSI being lower on INF cells relative to PRE cells). In scenario (iii), HIV preferentially infects cells with overall equivalent levels of the antigen (as reflected by the MSI being the same on PRE cells and UI cells) but then downregulates the antigen in INF cells (as reflected by the MSI being lower on INF cells than PRE cells). (B) F4.HSA downregulates CD28, ICOS, and CD45RO, components of the TCR signaling apparatus. *p<0.05, **p<0.01 as assessed using the Student's paired t test and adjusted for multiple testing using the Benjamini-Hochberg for FDR. (C) F4.HSA downregulates the α4 component of the α4β1 integrin. *p<0.05, **p<0.01 as assessed using the Student's paired t test and adjusted for multiple testing using the Benjamini-Hochberg for FDR. (D) CCR7 expression is higher on Tm cells than on memory CD8+ T cells from the endometrium. *Left:* Histogram plots showing memory T cells gated on CD4 or CD8 as indicated. *Right:* Summary of data from 8 donors. ****p<0.0001 as assessed using the Student's paired t test. (E) F4.HSA preferentially infects Tm cells with low levels of CCR7 but then upregulates expression of this receptor. *Left:* Plot showing MSI of CCR7 on UI, PRE, and INF cells. *Right:* Plot showing higher levels of expression of the HIV co-receptor CCR5 on CCR7- Tm cells, which provides an explanation for the higher susceptibility of these cells to infection. *p<0.05 as assessed using the Student's paired t test. (F) F4.HSA upregulates expression of follicle-homing receptor CXCR5 and the lymph node retention marker CD69. **p<0.01, ***p<0.001 as assessed using the Student's paired t test and adjusted for multiple testing using the Benjamini-Hochberg for FDR.

The online version of this article includes the following figure supplement(s) for figure 5:

**Figure supplement 1.** PP-SLIDE successfully identifies CD4+ T cells as preferential T cell targets of infection and detects downregulation of CD4 upon infection.

magnitude higher than that in unstimulated PBMCs, it was still lower than that in unstimulated ETs (*Figure 2—figure supplement 7C*). PP-SLIDE analysis revealed that HIV induced remodeling in all 3 specimens with Tm as the major targets (*Figure 2—figure supplement 7D,E*). However, PRE cells from the endometrium and stimulated PBMCs resided in distinct regions of the t-SNE plot, suggesting that these two types of cells, although both highly permissive to HIV, are phenotypically distinct. The activation markers CD25, HLA-DR, CD28, ICOS, and Ox40 were all lower on the PRE cells from ETs than from activated PBMC T cells (*Figure 2—figure supplement 7F*). These data demonstrate that ETs are less activated than stimulated PBMCs, yet more permissive to HIV infection, suggesting that activation state does not fully account for the high permissivity of ETs to infection. In contrast, CCR5 expression levels were higher in the ETs than stimulated PBMCs (*Figure 2—figure supplement 7G*), suggesting that the high permissivity of ETs to infection may be in large part due to their high expression of the HIV co-receptor.

## HIV preferentially infects Th1, Th2, Tfh, and Trm ETs

We next determined to what extent the HIV-susceptible ETs were enriched among the canonical subsets of differentiated CD4+ T cells. F4.HSA-infected ETs from 4 additional donors, for a total of 8, were analyzed by CyTOF to ensure a more robust statistical analysis. The new datasets were subjected to PP-SLIDE to identify the PRE cells (*Figure 4—figure supplement 1*). Manual gating was used to determine the percentages of Tem, Tcm, Th1, Th2, Th17, regulatory T cells (Treg), T follicular helper cells (Tfh), and Trm cells among the PRE versus Tm cells. Consistent with the data presented earlier, Tem cells were preferentially infected and Tcm cells were selectively spared. Th1, Th2, Tfh, and Trm cells were preferentially infected, while there was no significant difference in the proportions of Th17 or Treg cells between PRE and uninfected Tm cells (*Figure 4*). The preferential infection of Th1 and Th2 but not Th17 cells may be in part due to the relatively low frequencies of Th17 cells in the endometrium (*Figure 4—figure supplement 2A*). Of note, the vast majority of Tem, Th1, Th2, and Th17 cells expressed the Trm marker CD69, consistent with these differentiated cells establishing residence in the endometrium (*Figure 4—figure supplement 2B*).

## HIV remodels genital T cells to impair TCR signaling and promote dissemination of infected cells

Having described the phenotypes of HIV-susceptible ETs, we next assessed how HIV modulates expression of specific receptors on these cells during infection by comparing antigen expression in PRE versus infected cells (*Figure 5A*). For example, an antigen that is expressed at higher levels on cells selected for infection would exhibit a higher median signal intensity (MSI) on PRE cells than on uninfected cells (*Figure 5Ai*, 'Selection Step'). If infection does not modulate expression of that antigen, its MSI would be the same in the PRE and infected cells (*Figure 5Ai*, 'Remodeling Step'). If, however, the antigen is downregulated as a result of infection, then its MSI will be lower on infected cells relative to PRE cells (*Figure 5Aii*). *Figure 5Aiii* shows an additional hypothetical example of an antigen whose expression levels are equivalent on the total population of uninfected cells and on cells preferentially infected by HIV, but are then downregulated as a result of infection.

Using such an approach, we first confirmed the expected pattern of CD4 expression. PP-SLIDE revealed, as expected, that PRE cells had a higher CD4 MSI than the total population of uninfected T cells, suggesting that HIV selects cells expressing high levels of CD4 for infection. The PRE cells also had higher CD4 MSI than the infected cells, consistent with downregulation of CD4 upon infection (*Figure 5—figure supplement 1*). We then explored whether other components of the TCR signaling apparatus were also modulated by infection. CD28 and ICOS, co-stimulatory receptors important for signaling through the TCR, were both downregulated by infection (*Figure 5B*). Interestingly, the levels of CD45RO, a splice form of CD45 which is important in T cell activation and stabilizing T cell binding to B cells (*Ledbetter et al., 1993*; *Stamenkovic et al., 1991*), was also downregulated (*Figure 5B*). These results demonstrate that multiple components of the TCR signaling apparatus are downregulated upon HIV infection of genital T cells.

We went on to study whether any modulations that might promote systemic spread of the virus could be identified. The integrin α4β1 directs CD4+ T cells to the FRT (*Davila et al., 2014*). Although we did not see differences in β1 expression between uninfected, PRE, or infected cells,

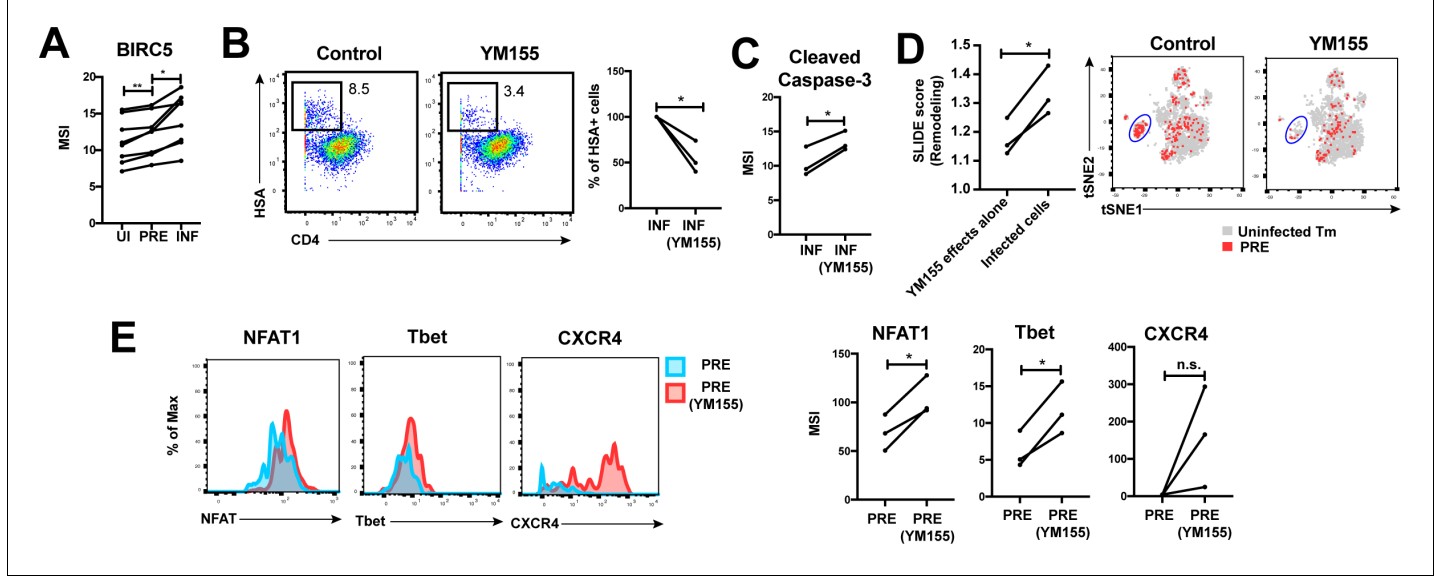

**Figure 6.** HIV exploits BIRC5 for cell survival upon infection of genital T cells. (**A**) F4.HSA preferentially infects Tm cells with high levels of BIRC5, and then further upregulates expression of this protein during infection. *p<0.05, **p<0.01 as assessed using the Student's paired t test. (**B**) *Left:* Representative plots of CyTOF analysis of the frequencies of HIV-infected cells in the absence vs. presence of the BIRC5 inhibitor YM155 (200 nM). Results are gated on live, singlet Tm cells. *Right:* Data from 3 donors (Donors 11–13, *Supplementary file 1*) demonstrating significant decrease in proportion of HIV-infected cells upon YM155 treatment. (**C**) HIV-infected cells that persist in the presence of YM155 are more pro-apoptotic as assessed by higher expression levels of cleaved caspase-3 relative to infected cells that were never exposed to YM155. (**D**) *Left:* SLIDE analysis revealing significantly higher remodeling in the productively-infected YM155-treated cells than in the uninfected YM155-treated cells. *p<0.05, **p<0.01 as assessed using the Student's paired t test. *Right:* YM155 promotes death of some infected Tm over others. t-SNE plots of uninfected (grey) and PRE (red) cells for cultures exposed or not to YM155. Highlighted in blue is a region of the t-SNE preferentially devoid of PRE cells in the YM155 sample, suggesting preferential killing of these cells upon infection. Data are representative of a total of 3 independent donors. (**E**) Antigens differentially expressed on HIV-susceptible cells that survive infection in the presence of YM155. PRE cells from YM155-exposed cultures were compared to those from cultures never exposed to the drug for expression levels of the indicated antigens. *Left:* Representative histogram plots. *Right:* Cumulative data from 3 donors. *p<0.05, as assessed using the Student's paired t test. n.s.: non-significant.

there was selection of $\alpha 4^{low}$ cells for infection, followed by further down-regulation of $\alpha 4$ after infection (*Figure 5C*). These events may facilitate exit of the infected cells from the genital tract.

Because lymph nodes provide a rich source of permissive CD4+ T cells that can propagate viral spread, we assessed whether there was evidence of upregulation of homing receptors that promote migration to lymph nodes. CCR7 is a homing receptor whose cell-intrinsic expression has been shown to be necessary for CD4+ T cells to depart mucosal tissues and migrate into draining lymph nodes (*Bromley et al., 2013*). Recent studies using cells isolated from cervicovaginal lavages (CVLs) demonstrated that Tm cells express high levels of CCR7 relative to memory CD8+ T cells, and that ex vivo exposure of these cells to HIV resulted in equivalent numbers of infected CCR7+ and CCR7-Tm cells (*Swaims-Kohlmeier et al., 2016*). We found that similar to cells from CVLs, endometrial Tm expressed more CCR7 than their CD8+ counterparts and that CCR7 levels were similar between uninfected and HIV-infected cells (*Figure 5D and E*). Interestingly, however, infected cells expressed higher levels of CCR7 than PRE cells (*Figure 5E*). This finding suggests that cells with low levels of CCR7 are preferentially susceptible to infection, but that HIV infection actively upregulates expression of CCR7, which may help direct these cells to the lymph nodes. The higher susceptibility of the CCR7- Tm cells to HIV infection is consistent with the notion that Tem (which are CCR7-) are preferentially targeted for infection (*Figure 4*) and may be attributable to the relatively higher levels of CCR5 on these cells relative to their CCR7+ counterparts (*Figure 5E*).

Given that HIV actively replicates within the follicles of lymph nodes (*Cavrois et al., 2017*), we also assessed expression levels of CXCR5, a chemokine receptor that directs cells from the extrafollicular region to the lymphoid follicles. CXCR5 was preferentially expressed on PRE cells, consistent

with Tfh (which are CXCR5+) being preferential HIV targets (*Cavrois et al., 2017*; *Figure 4*), and further upregulated by HIV after infection (*Figure 5F*). CD69, which retains cells within lymph nodes (*Baeyens et al., 2015*), was also markedly upregulated by HIV infection (*Figure 5F*). Together, these data demonstrate that surface expression of both chemokine receptors and integrins are altered by HIV infection in ways that may promote viral dissemination to the lymphoid follicles of lymph nodes.

## HIV promotes survival of infected genital T cells by upregulating BIRC5

Finally, we postulated that for effective dissemination, the infected cell needs to survive for long enough to travel the distance to infect other cells. BIRC5 was recently shown to promote survival of HIV-infected PBMCs (*Kuo et al., 2018*). We found that HIV targeted cells in the FRT with high levels of BIRC5 for infection, and then further upregulated this protein (*Figure 6A*). These results highlight that HIV-infected genital T cells, like infected cells from PBMCs (*Kuo et al., 2018*), express high levels of BIRC5, but importantly suggest that this reflects a combination of HIV selecting cells for high levels of BIRC5 and upregulating this anti-apoptotic factor.

To further assess whether BIRC5 activity in genital T cells is targetable, we used CyTOF to analyze the effect of YM155, a clinically-tested pharmacological inhibitor of BIRC5 (*Clemens et al., 2015*), on HIV infection of ETs from 3 new donors (*Supplementary file 1*). Treatment of ETs with YM155 reduced the frequency of cells infected by F4.HSA (*Figure 6B*), similar to what was observed with PBMC-derived cells (*Kuo et al., 2018*). Furthermore, the cells that became infected despite the presence of YM155 expressed higher levels of cleaved caspase-3, a hallmark of apoptotic cells (*Figure 6C*). These cells were remodeled, and their remodeling was driven by productive infection and not by effects of YM155 on cell phenotypes (*Figure 6D*, left). Interestingly, PRE cells from YM155-treated specimens were distributed differently than PRE cells from cultures not exposed to the drug, suggesting that YM155 preferentially targets a subset of Tm cells for destruction (*Figure 6D*, right). The PRE cells that survived infection despite the presence of YM155 expressed relatively high levels of NFAT, Tbet, and CXCR4 (*Figure 6E*). These data suggest that BIRC5 is important for survival of some but not all HIV-infected genital T cells.

Collectively, these studies of remodeling support a model whereby HIV infection of genital T cells generates a population of infected Tem that exhibit enhanced survival due to BIRC5 expression but compromised TCR signaling. Based on the expression patterns of homing receptors, these infected cells are expected to disseminate to the lymphoid follicles of lymph nodes (*Figure 7*).

## Discussion

In this study, we use CyTOF and high-dimensional analytical approaches to define the features of the most HIV-susceptible cells present in the endometrium of the FRT. We further explore how these cells are remodeled by HIV and the potential biological consequences of these changes. We find that endometrial CD4+ T cells undergo unusually high levels of productive infection with R5-tropic HIV-1, likely reflecting in part the high expression of the CCR5 co-receptor on these cells. Consistent with our results, a prior study had found higher levels of HIV fusion to CD4+ T cells from FRT compared to blood (*Joag et al., 2016*). However, these fusion assays

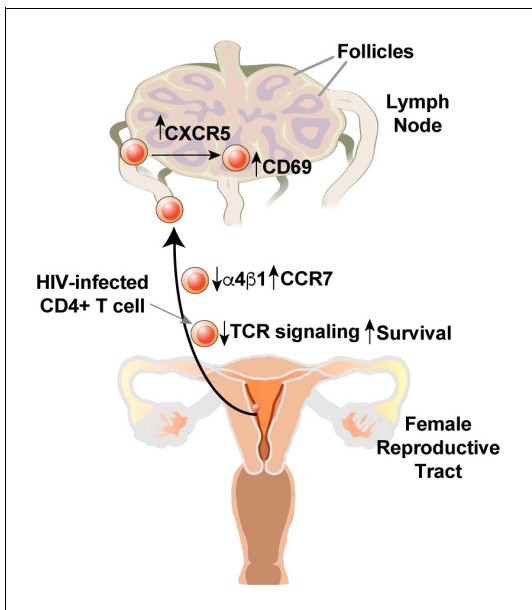

**Figure 7.** Model of how HIV-induced remodeling of genital T cells can promote systemic spread of the virus. Infected cells from the genital tract are poorly capable of mounting an effective immune response due to downregulation of multiple components of the TCR signaling apparatus, and survive due to upregulation of BIRC5. Downregulation of α4β1 and upregulation of CCR7 promote migration of the infected cells from the genital mucosa to draining lymph nodes, where upregulation of CXCR5 and CD69 promote migration to and retention within the follicles.

do not provide information on remodeling since these events occur later in the viral life cycle. Additionally, fusion does not equate productive infection due to the presence of various post-entry host restriction factors for HIV. In our study, we have characterized productively-infected cells, and by converting all infected cells to PRE cells via PP-SLIDE, were able to conduct an in-depth phenotypic analysis of the types of genital T cells preferentially targeted for productive infection by HIV, and to define precisely the remodeling events that occur in these cells.

The preferential targets of HIV in the FRT were memory CD4+ T cells, while their naïve counterparts were markedly resistant. Moreover, the most susceptible cells corresponded to Tem cells, while Tcm cells were largely spared. We observed preferential infection of Tem in PBMCs as well, consistent with prior reports of blood Tem cells being more susceptible than Tcm to DC-mediated *trans*-infection by an R5-tropic HIV (*Groot et al., 2006*). Further characterization of the highly-susceptible Tem cells from the FRT revealed them to be comprised of multiple effector subsets including, Th1, Th2, Th17, Treg, Tfh, and Trm cells, with the Th1, Th2, Tfh, and Trm cell subsets being particularly prone to infection. These subsets have been reported to also be preferentially targeted by HIV in blood and lymph nodes (*Gosselin et al., 2010*; *Moonis et al., 2001*; *Perreau et al., 2013*; *Zhang et al., 2012*). In addition, Trm cells were previously identified as preferential targets for infection in the cervix (*Cantero-Pérez et al., 2019*), similar to our findings in endometrial cells. Surprisingly, although we found Th17 cells among the HIV-susceptible cells in the endometrium, we did not find evidence of preferential selection of this cellular subset. This result differed from the observed increased infection of genital Th17 cells in SIV-infected macaques (*Stieh et al., 2014*). The reason for the apparent discrepancy is unclear, but may reflect differences between HIV and SIV, tissue site examined, conditions of infection, and/or the relatively low frequencies of Th17 in the human endometrium. Of note, the preferential infection of Th17 cells by SIV was observed primarily in the lower reproductive tract with lower infection levels in the upper FRT (*Stieh et al., 2014*). Consistent with our results, an ex vivo HIV infection study did not find a significant difference in the proportion of infected endometrial T cells that did or did not express the Th17 marker CCR6. This study also found a relatively low frequency of Th17 cells in the endometrium relative to the lower FRT (*Rodriguez-Garcia et al., 2014*).

Our study also led to important insights into how HIV remodels its host cell. Interestingly, the remodeling that was observed required productive infection, since bystander Tm cells in infected cultures exhibited limited remodeling, as did Tm cells treated with culture supernatants from infected cultures. Cell-intrinsic innate immune recognition, active remodeling by viral accessory genes, and/or DNA damage responses as a result of viral integration are all potential contributors to the observed remodeling. Which proteins exhibit altered expression levels as a result of infection? Prior SILAC-based proteomics have been used to address this question but due to sample requirements, these analyses were limited to cell lines (*Matheson et al., 2015*). By implementing PP-SLIDE on HIV-infected genital cells phenotyped by CyTOF, we were able to characterize cellular remodeling by HIV at the single-cell level. We find that HIV significantly remodels both endometrial cells and PBMCs following infection and that remodeling involves downregulation of cell-surface CD4, as expected (*Garcia and Miller, 1991*; *Vincent et al., 1993*), but also downregulation of other components of the TCR complex. In particular, the co-stimulatory molecules CD28 and ICOS, and the CD45RO spliced from of the CD45 phosphatase, are significantly downregulated by HIV infection. CD28 downregulation has been linked to Nef activity and is thought to promote disengagement of T cells from antigen presenting cells, thereby disrupting TCR signal transduction (*Cavrois et al., 2017*; *Swigut et al., 2001*). ICOS is closely related to CD28 and the two function similarly during expansion, survival, and differentiation of T cells (*van Berkel and Oosterwegel, 2006*). The downregulation of both co-stimulatory molecules by HIV, along with CD45 that stabilizes the interactions of T cells with B cells (*Ledbetter et al., 1993*), suggests that disrupting TCR signaling in CD4+ T cells at the portal of viral entry may provide an evolutionary advantage for the virus.

We also identify a number of host proteins that are upregulated by HIV during infection, including CCR7, CXCR5, and CD69. A recent study suggested that CCR7 may drive migration of HIV-infected cells from the lower FRT to the lymphatics (*Swaims-Kohlmeier et al., 2016*). That study also demonstrated that HIV-infected T cells from CVL consist of equal proportions of CCR7+ and CCR7- Tm cells (*Swaims-Kohlmeier et al., 2016*). Consistent with these findings, we find that the levels of CCR7 are not significantly different in uninfected and infected Tm cells. Interestingly, we do find that PRE cells express significantly lower levels of CCR7 than both total Tm cells and HIV-infected cells,

suggesting that HIV selects CCR7$^{low}$ cells for infection, but then as part of remodeling upregulates expression of this homing receptor. The likely reason for preferential infection of the CCR7$^{low}$ cells is high expression of the CCR5 co-receptor on these cells. The reason behind why HIV has evolved to upregulate CCR7 is unclear, but we speculate that it may help promote systemic spread of the virus from the FRT by directing infected cells to regional lymph nodes.

Further supporting this notion is our finding that the virus also upregulates expression of CXCR5, which directs T cells into the lymph node follicles, a known site of HIV replication in infected individuals (*Cavrois et al., 2017*). Together with our observation and those of others (*Matheson et al., 2015*) that HIV downregulates the α4 component of the genital-homing α4β1 integrin, these findings highlight how infected cells from the genital tract may be directed to the follicles of draining lymph nodes where a rich source of CD4+ target cells can further facilitate even broader tissue spread of the virus.

The finding that HIV preferentially infects CD69+ Trm cells from the FRT (*Cantero-Pérez et al., 2019*) may seem at odds with our model for how infection promotes changes leading to viral dissemination, since CD69 inhibits S1P1 to promote cellular retention within tissues (*Bankovich et al., 2010*). Furthermore, we observed that HIV even further upregulated CD69 upon infection, similar to what has been observed in a cell line (*Matheson et al., 2015*). However, it is important to note that CD69 expression is not sufficient to confer tissue residence (*Beura et al., 2018*) and that CD4+ Trm cells tend to circulate to a greater exent than their CD8+ counterparts (*Beura et al., 2019*; *Collins et al., 2016*; *Gebhardt et al., 2011*). We postulate that a subpopulation of infected cells expressing CD69, CCR7, and CXCR5 continue to traffic from the genital mucosa to the follicles of regional lymph nodes promoting viral spread.

In addition to remodeling of homing receptors, HIV infection of genital T cells also upregulates expression of BIRC5, an anti-apoptotic protein recently shown to promote the survival of HIV-infected blood-derived CD4+ T cells (*Kuo et al., 2018*). Interestingly, PP-SLIDE suggests that the increase in BIRC5 expression in HIV-infected genital T cells is due to a combination of HIV preferentially infecting cells with higher levels of BIRC5 followed by further upregulation of this protein within the infected cells. The mechanism by which HIV actively upregulates BIRC5 expression remains unclear, but these events likely help further propel viral dissemination. Inhibition of BIRC5 with the YM155 small molecule antagonist is associated with decreased frequency of HIV-infected genital cells likely reflecting increased apoptosis of these cells. Interestingly, the HIV-susceptible cells that survive in the presence of YM155 exhibit unique phenotypic features including increased levels of NFAT, Tbet, and CXCR4. These factors have been previously associated with inhibition of apoptosis and promotion of T cell survival and may compensate for the loss of BIRC5 function (*Lee et al., 2018*; *Suzuki et al., 2001*). Defining the precise signaling pathways enabling survival of these remaining cells may help identify additional drug targets aimed at limiting the survival of HIV-infected cells.

Our experimental system has some limitations. Like any system that entails viral infection of primary cells, it requires that the cells be cultured for multiple days in order to allow enough time for productive infection to occur. The ex vivo culture of primary cells can lead to phenotypic drift, the phenomenon whereby the phenotypes of cells diverge from their original in vivo state during ex vivo culture. A second limitation of our system is that the endometrial biopsies need to be digested with collagenase to isolate single-cell suspensions of FRT cells for the infection assay. This digestion cleaves some cell-surface markers, although re-expression of the markers occurs upon culturing of the cells. While these effects are impossible to avoid, we do not think they biased our results as all our infected endometrial specimens were compared to mock-treated cells that were digested similarly and cultured for the same period of time.

Our findings that HIV infects a diverse array of CD4+ T cells in the FRT (with the notable exception of naïve and Tcm cells), and remodels these cells in ways that can promote viral dissemination, may help explain the generally poor effectiveness of oral PrEP or microbicides in protecting women from sexual transmission of HIV. However, certainly other factors including adherence and genital drug levels also play important roles (*Cottrell et al., 2016*; *Hodges-Mameletzis et al., 2019*; *Riddell et al., 2018*). Our findings also suggest that vaccines designed to elicit protective anti-HIV CD4+ T cell responses in the genital mucosa, for example via the 'prime and pull' strategy (*Bernstein et al., 2019*; *Shin and Iwasaki, 2012*), need to consider the high diversity of HIV-susceptible genital T cells present and the high levels of HIV infection supported by these cells. Such

strategies may perhaps be most effective if they can preferentially elicit the poorly-susceptible subsets of Tcm and encourage their migration into the FRT. Moreover, combination microbicides that include components that can either disrupt the lymph node homing affinity of HIV-infected cells or the ability of HIV to promote the survival of infected cells – in particular the clinically approved BIRC5 inhibitor YM155 – could improve the overall effectiveness of these agents and provide at-risk women with the power to protect themselves from HIV acquisition. A key next step for the field will be to confirm in vivo the findings reported here using FRT specimens from non-human primate models of HIV transmission or from people living with HIV.

## Materials and methods

### Endometrial tissue processing and cell isolation

Fresh human endometrial biopsies from HIV seronegative women were obtained from the University of California, San Francisco (UCSF) (IRB # 14–15361) and the Women's Health Clinic of Naval Medical Center Portsmouth (NMCP) in Virginia (CIP # NMCP.2016.0068), under standard operating procedures (Fassbender et al., 2014). Donor information (age, race, cycle day, gynecological diagnosis, pathology, and progesterone levels) associated with the biopsies is summarized in Supplementary file 1. The biopsies from NMCP were delivered in MCDB-105 (Sigma-Aldrich M6395) supplemented with 10% heat-inactivated fetal bovine (FBS) and 1% Penicillin/Streptomycin (P/S) and processed within 48 hr following surgery; biopsies from UCSF Endometrial Tissue Bank were processed within 6 hr after surgery. To isolate leukocytes from the biopsies, specimens were washed with SCM media, consisting of 75% phenol red-free Dulbecco's Modified Eagle's Medium (DMEM, Life Technologies) and 25% MCDB-105 supplemented with 10% charcoal-stripped fetal bovine serum, 1% L-glutamine with P/S (Gemini), 1 mM sodium pyruvate (Sigma-Aldrich), and 5 mg/ml insulin (Sigma-Aldrich). The specimens were then digested for 2 hr under rotation at 37°C in SCM medium mixed at a 1:1 dilution of Digestion Media. The Digestion Media consisted of HBSS containing $Ca^{2+}$ and $Mg^{2+}$ supplemented with 3.4 mg/ml collagenase type 1 (Worthington Biochemical Corporation LS004196) and 100 U/ml hyaluronidase (Sigma-Aldrich H3631). Digested cells were then filtered through a Falcon 40 μm cell strainer and the filtrate was washed with R10 media (RPMI 1640 containing 10% FBS and 1% P/S). The cells were then cultured in 96-well U-bottomed polystyrene plates at a concentration of $10^6$ cells / well in 200 μl. Of note, endometrial cells were never stimulated with a mitogen at any point during in vitro culture.

### PBMC isolation

PBMCs were isolated from reduction chambers obtained from Vitalant Research Institute or the NMCP. All blood products were first diluted at a 1:1 ratio using FACS buffer (PBS supplemented with 2% FBS and 2 mM EDTA). Ficoll (Stemcell Technologies) was then slowly added at the bottom of the tube at a ratio of 2:1. The samples were then centrifuged at 931 g for 30 min. After centrifugation, the PBMC layer underlying the top supernatant was transferred to a new tube and cells were washed 3x in FACS buffer. Where indicated, PBMCs were activated by stimulation for 3 days with 5 μg/ml PHA (Sigma-Aldrich) in the presence of 100 IU/ml IL-2 (Life Technologies). Activated PBMCs were always maintained in R10 media containing 20 IU/ml IL-2. For sorting experiments, memory CD4+ T cells (Tm cells) were purified using the EasySep CD4 negative selection enrichment kit (Stem cell Technologies) followed by depletion of naïve T cells using CD45RA beads (Miltenyi Biotec) prior to antibody staining.

### Generation of human lymphoid aggregate cultures (HLAC) from tonsils

Human tonsils were obtained from the Cooperative Human Tissue Network (CHTN). To generate HLACs, tonsils were dissected into ~1 $cm^2$ pieces and pushed through a 40 μm cell strainer using a syringe plunger. The cells were then filtered through a second 40 μm cell strainer and cultured in 96-well U-bottomed polystyrene plates at a concentration of $10^6$ cells / well in 200 μl tonsil media (RPMI supplemented with 15% FBS, 100 μg/ml gentamicin, 200 μg / ml ampicillin, 1 mM sodium pyruvate, 1% non-essential amino acids (Mediatech), 1% Glutamax (Thermo Fisher), and 1% Fungizone (Invitrogen)).

## Virus preparation and infection assays

The F4.HSA HIV DNA construct has been described previously (*Cavrois et al., 2017*). To generate F4.HSA stocks, 293 T cells were transfected by polyethylenimine (PEI)-mediated transfection (Polysciences) with F4.HSA proviral DNA (70 μg / flask) (*Longo et al., 2013*). At 2 days post-transfection, 293T supernatants containing virus were filtered through a 0.22 μm filter and concentrated by ultracentrifugation at 20,000 rpm (Beckman Coulter Optima XE-90) for 2 hr at 4°C. Viral titers were quantitated using the Lenti-X p24$^{Gag}$ Rapid Titer Kit (Clontech). PBMCs (unstimulated or PHA-stimulated), unstimulated tonsillar HLACs, or unstimulated endometrial cells were inoculated with the viral stocks. Because recent studies demonstrated that fibril-based enhancers of HIV infection, including those naturally present in human semen (*Roan et al., 2014*; *Roan et al., 2011*; *Usmani et al., 2014*; *Yolamanova et al., 2013*), can boost HIV infection rates, we tested to what extent fibrils EF-C (*Yolamanova et al., 2013*) and SEM86 (*Roan et al., 2014*) increase F4.HSA infection of target cells. A total of 100–200 ng/ml p24$^{Gag}$ of F4.HSA was pre-treated for 15–30 min with 50 μg/ml EF-C or 100 μg/ml SEM86, and then diluted 10-fold upon addition to the indicated cell type in 96-well U-bottomed polystyrene plates ($10^6$ cells in a final volume of 200 μl / well). This corresponded to a final p24$^{Gag}$ concentration of 10–20 ng/ml. Infection was then allowed to proceed for 2 hr, after which cultures were replaced with fresh media and allowed to incubate for another 3 days. Control cells were incubated with media for 3 days and harvested at the same time as the infected culture. Both EF-C and SEM86 markedly enhanced infection by F4.HSA (*Figure 1—figure supplement 1*). Although enhancement was more pronounced with EF-C than SEM86, we carried out all infection experiments with SEM86 as it is present in semen and its endogenous levels have been shown to significantly correlate with the infection-promoting activity of semen (*Roan et al., 2014*; *Usmani et al., 2014*), making it an appropriate reagent for characterizing sexual transmission by HIV, which typically occurs in the presence of semen. In experiments where the extent of remodeling was assessed in the presence of ART, supernatants from uninfected and F4.HSA-infected cultures were collected three days post-infection as a source of conditioned media. Uninfected cells isolated from the same specimen were pre-treated for 30 min in media containing a cocktail of suppressive ART (5 μM AZT, 5 μM Ritonavir, 8 nM Efavirenz, 10 μM Lamivudine, 50 nM Raltegravir, and 0.5 μg/ml T-20 (all from NIH AIDS reagent program)), spun down, and then resuspended in the conditioned media in the presence of the ART cocktail. The cultures were allowed to incubate for another two days, after which cells were harvested for CyTOF analysis. Where indicated, cells were infected in the presence of 200 nM YM155 (Selleckchem) and after 2 hr replaced with fresh media containing 200 nM YM155.

## Flow cytometry

For each sample, 0.1–1 million cells were transferred into 96-well V-bottom polystyrene plates and washed once with FACS buffer, stained for 30 min on ice with APC/Cy7-CD3 (SK7, Biolegend), PE/Cy7-CD4 (A161A1, Biolegend), APC-CD8 (SK1, Biolegend), FITC-CD24 (HSA, M1/69, BD Biosciences), and the LIVE/DEAD Zombie Aqua Fixable Viability Kit (Biolegend). After 3 additional washes with FACS buffer, cells were analyzed by flow cytometry on an LSRFortessa or LSR II (BD Biosciences).

For sorting experiments, purified Tm cells were washed once with FACS buffer and stained at room temperature for 15 min with a 1:200 dilution of the LIVE/DEAD Zombie Aqua Fixable Viability Kit (Biolegend). After washing 1x with FACS buffer, cells were stained for 30 min at 4°C with a cocktail of antibodies diluted in a 1:1 mixture of FACS buffer and the Brilliant Stain Buffer (BD Biosciences). The antibody cocktail consisted of APC/Cy7-CD3 (SK7, Biolegend), BD Horizon BV650-CD8 (RPA-T8, BD Biosciences), BD Horizon BUV737-CD4 (SK3, BD Biosciences), FITC-CD45RO (UCHL1, Biolegend), BD OptiBuild BUV395-CD45RA (HI100, BD Biosciences), and PE/Dazzle 594-CD197 (CCR7, G043H7, Biolegend). The cells were then washed once in FACS buffer, resuspended at a concentration of 10 million cells / ml, and sorted using an AriaII flow cytometer (BD Biosciences). Fluorescence minus one (FMO) controls for CCR7 staining and compensation controls using Comp Beads (BD Biosciences) were acquired at the same time. Sorted CCR7$^{low}$, CCR7$^{medium}$, and CCR7$^{high}$ populations, along with total Tm, were infected with F4.HSA for 3 days and then analyzed by FACS as described above.

## Development of a CyTOF panel to characterize fixed endometrial T cells

For this study, we designed a 38-parameter CyTOF panel that includes markers of T cell differentiation states, activation markers, transcription factors, and homing receptors, along with an antibody against HSA to identify productively-infected cells (*Supplementary file 2*). Antibodies that required in-house conjugation were conjugated to their corresponding metal isotopes using X8 antibody labeling kits according to manufacturer's instructions (Fluidigm). Briefly, metal was loaded onto polymer and incubated for 1 hr at room temperature. Unconjugated antibody was transferred into 50 kDa Amicon Ultra 500 µl V-bottom filters (Fisher), and reduced for 30 min at 37°C using a 1:125 dilution of TCEP (Pierce). After 2 washes with C-buffer (Fluidigm), metal-loaded polymer was resuspended in 200 µl C-buffer within 3 kDa Amicon Ultra 500 µl V-bottom filters (Fisher). Metal-loaded polymer was then transferred to the appropriate antibody and coupled for 2 hr at 37°C. After 3 washes with W-buffer (Fluidigm), the conjugated antibodies were quantitated for protein content by Nanodrop, diluted 1:1 using a PBS-based Antibody Stabilizer (Boca Scientific) supplemented with 0.05% sodium azide, and stored at 4°C.

For live/dead discrimination, 1–6 million cells were washed 1x with contaminant-free PBS (Rockland) supplemented with 2 mM EDTA (PBS/EDTA), incubated for 60 s at room temperature with 25 µM cisplatin in 4 ml PBS/EDTA, and then quenched with CyFACS (metal contaminant-free PBS (Rockland) supplemented with 0.1% bovine serum albumin and 0.1% sodium azide). Cells were then fixed with 2% PFA in metal contaminant-free PBS (Rockland), washed 3x with CyFACS, and frozen at −80°C until CyTOF staining.

To stain multiple samples in the same reaction, barcoding of cells was conducted using the Cell-ID 20-Plex Pd Barcoding Kit according to manufacturer's instructions (Fluidigm). Briefly, for each sample, 1–3 million cells were washed 2x with Barcode Perm buffer (Fluidigm), and then the appropriate barcode was added at a 1:90 ratio and incubated with the cells for 30 min. Cells were then washed with 0.8 ml Maxpar Cell Staining buffer (Fluidigm), followed by with CyFACS. Barcoded samples were then combined and blocked for 15 min on ice with sera from mouse (Thermo Fisher), rat (Thermo Fisher), and human (AB serum, Sigma-Aldrich) in Nunc 96 Deep-Well polystyrene plates (Thermo Fisher). Cells were then washed 2x with CyFACS buffer, and stained for 45 min on ice with the cocktail of CyTOF surface staining antibodies (*Supplementary file 2*) in a total volume of 100 µl / well. Cells were then washed 3X with CyFACS buffer and fixed overnight at 4°C with 2% PFA (Electron Microscopy Sciences) in metal contaminant-free PBS (Rockland). The next day, cells were permeabilized by incubation for 30 min at 4°C with fix/perm buffer (eBioscience), and then washed 2x with Permeabilization Buffer (eBioscience). Cells were then blocked for 15 min on ice with sera from mouse (Thermo Fisher) and rat (Thermo Fisher). Cells were then washed 2x with Permeabilization Buffer (eBioscience), and stained for 45 min on ice with the cocktail of CyTOF intracellular staining antibodies (*Supplementary file 2*). Cells were then washed 1x with CyFACS and incubated for 20 min at room temperature with 250 nM Cell-ID DNA Intercalator-Ir (Fluidigm) in 2% PFA diluted in PBS. Cells were then washed 2x with CyFACS, 1x with Maxpar Cell Staining Buffer (Fluidigm), 1x with Maxpar PBS (Fluidigm), and 1x with Maxpar Cell Acquisition Solution (Fluidigm). Immediately prior to acquisition, cells were resuspended to a concentration of $7 \times 10^5$/ ml in EQ calibration beads (Fluidigm) diluted in Maxpar Cell Acquisition Solution. Cells were acquired at a rate of 250–350 events/sec on a CyTOF2 instrument (Fluidigm) at the UCSF Flow Cytometry Core.

The new CyTOF panel was validated on paraformaldehyde (PFA)-fixed cells by confirming expected expression patterns of each antigen on different subsets of cells (*Figure 2—figure supplement 8*). The ability to phenotype PFA-fixed cells overcame technical challenges associated with working with ETs, as it enabled barcoding samples from multiple donors into one batch, which not only limited batch effects but also increased the numbers of cells processed at a time, minimizing cell loss during the processing steps leading up to CyTOF analysis.

## CyTOF data analysis

Data were normalized to EQ calibration beads and de-barcoded with CyTOF software (Fluidigm). Normalized data were imported into both FlowJo (BD) and Cytobank for gating and further analysis. To identify total T cells, sequential gating was used based on DNA content, viability, cell length, and a CD3+CD19- gate (*Figure 2—figure supplement 1*). t-SNE plots were generated in Cytobank with

default settings and further analyzed in FlowJo. t-SNE plots were generated using on all parameters, except parameters that were used upstream in the gating strategy including HSA. For mapping of defined populations onto t-SNE plots, subsets were defined based on standard two-dimensional dot plots and then pseudo-colored on the t-SNE plot.

Identification of predicted precursor (PRE) cells by PP-SLIDE has been previously described (*Cavrois et al., 2017*). In our current analysis, a similar protocol was used to identify the uninfected nearest neighbor cell for each HIV-infected cell. The steps implemented are summarized below:

## Data cleanup and standardization

CD3+CD19- T cells from the uninfected culture and HIV-infected cells from the infected culture were gated on and exported using Flowjo10. To prepare for PP-SLIDE, the following parameters, which do not contain useful information for identifying the original cell type, were removed from the analysis:

| | |
|---|---|
| Non-informative markers | Live/dead staining, event length, barcodes, beads channel, DNA, time, background channel (190), and other non-cell markers |
| Infection marker | HSA |
| Marker highly modified by HIV infection and not informative for PRE analysis | CD4 |
| Markers used in upstream gating analysis | CD19, CD8 |

Raw expression values (signal intensity) of selected markers from each cell in the exported files were transformed by the inverse hyperbolic function (arcsinh) transformation as follows:

$$arsinh(x) = \ln\left( x + \sqrt{x^2 + 1} \right)$$

This transformation is used to standardize the diverse range of raw expression level scales for the measured parameters, and minimizes the effect of outliers and extreme numbers.

## Identification of PRE cell for each HIV-infected cell:

The Euclidean distance ($d_{F\_U}$) between each infected cell $F$ and each uninfected cell $U$ was calculated as follows:

$$d_{F\_U} = \sqrt{\sum_{i=1}^{n}(F_i - U_i)^2}$$

where n is the number of parameters analyzed and $i$ refers to the parameter being analyzed. For example, for parameter 1, $F_i - U_i$ would correspond to the value of parameter 1 on the infected cell minus the value of parameter 1 on the uninfected cell.

For each infected cell $F$, the $d_{F\_U}$ of all the uninfected cells U were sorted from lowest to highest to identify the shortest $d_{F\_U}$ value. This corresponds to the $k = 1$ nearest neighbor uninfected cell for that infected cell $F$, or the PRE cell. After identifying the PRE cells corresponding to each infected cell, the expression values corresponding to the original data matrix were exported as a new FCS file for downstream analysis. These PRE cells correspond to a subset of the original data matrix corresponding to total uninfected cells.

### FlowSOM analysis

FlowSOM is an unsupervised technique for clustering and dimensionality reduction (*Van Gassen et al., 2015*). FCS files corresponding to Tm cells (defined as CD3+CD8-CD45RA-CD45RO+ cells) were exported from FlowJo for FlowSOM analysis in Cytobank. FlowSOM was constructed using the default settings (Clustering method: Hierarchical consensus; Iterations: 10; Seed: Automatic) with a modification of total metaclusters from 10 to 20. All of the cell-related surface and intracellular antigens were used as clustering channels, except for CD19, CD8, CD45RA, and HSA, which were parameters used in the upstream gating strategy. For calculating the proportion of each cluster in

the pie chart, the total cell numbers and the cell numbers in each cluster were exported from Cyto-bank and the proportion of each cluster was calculated using the following equation:

P(X) = [(number of cells in Cluster X) / (total number of cells)] * 100

For each sample, the proportions of each cluster at a frequency >1% are shown (*Figure 3B* and *Figure 3—figure supplement 1B*).

To explore whether PRE cells preferentially resided within any of the clusters, or alternatively whether the PRE cells were stochastically distributed, we compared the frequency of each cluster among total uninfected Tm cells versus PRE cells by calculating chi-square values (see below). To examine the extent of enrichment of each cluster among PRE cells, we divided the frequency of each cluster among the PRE cells by the frequency of that same cluster among total uninfected Tm cells to obtain the ratio of cluster frequency:

Ratio(Cluster X) = Frequency(Cluster X among PRE) / Frequency(Cluster X among uninfected)

The ratios are reported in *Figure 3C* and *Figure 3—figure supplement 1C*. We designated clusters with ratios > 1 as enriched and those with ratios > 0 and <1 as non-enriched. Clusters labeled as 'undetectable' had lacked PRE cells within that cluster.

## Statistical analysis

The expression level of each parameter was reported by its median signal intensity (MSI). Student's two-sided paired (by donor) t-tests were used to test for differences in MSI of each parameter among uninfected cells, PRE cells, and infected cells, or among PRE cells from ETs and PBMCs. P-values were adjusted for multiple testing using False Discovery Rate (FDR) via the Benjamini-Hoch-berg method. FDR adjusted P-values that were less than 0.05 were considered as significant.

SLIDE analysis was conducted using the R package SLIDE (*Mukherjee et al., 2018*). The method tests the global hypothesis of remodeling in infected cells based on multi-parameter protein expression data (*Sen et al., 2014*). A nearest-neighbor approach is used in the SLIDE test, which enables it to identify remodeling in the presence of heterogeneity in the uninfected samples. Each infected cell is projected onto the uninfected sample to obtain its PRE cell. As there can wide variability in the characteristics of these PRE cells which can arise from different sub-populations, we compared the distance between the infected cell and its PRE cell taking into account the uninfected sample density at the PRE cell. This was done by calculating the distance between the PRE cell to its nearest neighbor cell in the uninfected sample. The ratios between these distances provides the remodeling score based on which the P-value of having significant viral remodeling of 20% or more global fold change is reported. To determine whether the SLIDE scores for infected PBMCs and ETs were significantly different, we considered a two-sample non-paired t-test based on the remodeling ratios from the two experiments corresponding to each particular donor, and reported the P value. Where indicated, SLIDE scores were calculated between bystander or conditioned media-treated cells by matching these cells against their uninfected cell counterparts, or among YM155-treated uninfected cells by matching against their non-YM155-treated uninfected counterparts.

Chi-squared test statistics values from the FlowSOM analyses were generated based on a goodness of fit test between the distribution of clusters between uninfected Tm and the PRE cells. To determine whether the chi-squared values between PBMCs and ETs were different, a conservative asymptotic test incorporating degrees of freedom correction was implemented. If $S_p(n_p)$ and $S_e(n_e)$ were respectively the chi-square statistic values from the PBMC and ET comparisons based on $n_p$ and $n_e$ groups, then, for testing the alternative hypothesis that PBMC chi-square values were significantly higher than the corresponding ET values, the P-value was computed based on $\frac{S_p(np) - S_e(ne) - np + ne}{\sqrt{2(np+ne-2)}}$ having an asymptotic standard normal distribution.

## Acknowledgements

This work was supported by the National Institutes of Health (R01 AI127219, R01 AI147777, and P01 AI131374 to NRR) and the National Science Foundation (NSF-DMS-1811866 to GM). We would also like to acknowledge the MACS/WIHS Combined Cohort Study (U01 HL146242); the DRC Center Grant P30 DK063720 and S10 1S10OD018040 for use of the CyTOF instrument; the S10 RR028962 for the sorter; and support from CFAR (P30 AI027763) and the Pendleton Foundation. We thank N Lazarus and E Butcher for the Act1 antibody. We thank S Tamaki and C Bisbo for CyTOF assistance

at the Parnassus Flow Core, N Raman for assistance in flow cytometry at the Gladstone Flow Core, G Maki for assistance in figure preparation; F Chanut for editorial assistance; and R Givens for administrative assistance.

## Additional information

### Funding

| Funder | Grant reference number | Author |
|---|---|---|
| National Institutes of Health | R01 AI127219 | Nadia Roan |
| National Institutes of Health | R01 AI147777 | Nadia Roan |
| National Institutes of Health | P01 AI131374 | Nadia Roan |
| National Science Foundation | NSF-DMS-1811866 | Gourab Mukherjee |
| National Institutes of Health | U01 HL146242 | Nadia R Roan |

The funders had no role in study design, data collection and interpretation, or the decision to submit the work for publication.

### Author contributions

Tongcui Ma, Conceptualization, Data curation, Formal analysis, Validation, Investigation, Visualization, Methodology, Writing - original draft, Writing - review and editing; Xiaoyu Luo, Resources, Software, Methodology; Ashley F George, Trimble L Spitzer, Linda C Giudice, Resources, Methodology; Gourab Mukherjee, Software, Formal analysis, Methodology; Nandini Sen, Conceptualization, Methodology; Warner C Greene, Conceptualization, Writing - review and editing; Nadia R Roan, Conceptualization, Resources, Data curation, Formal analysis, Supervision, Funding acquisition, Investigation, Methodology, Writing - original draft, Project administration, Writing - review and editing

### Author ORCIDs

Nadia R Roan (iD) https://orcid.org/0000-0002-5464-1976

### Decision letter and Author response

Decision letter https://doi.org/10.7554/eLife.55487.sa1
Author response https://doi.org/10.7554/eLife.55487.sa2

## Additional files

### Supplementary files

• Supplementary file 1. Clinical parameters of donor specimens analyzed by CyTOF. Table of patient clinical parameters (age, race, cycle day, diagnosis, cycle phase, progesterone levels, and specimen source) for each of the specimens analyzed by CyTOF.

• Supplementary file 2. List of CyTOF staining antibodies. Table showing the antigen, metal label, antibody clone, and source of the CytOF antibodies used in this study.

• Supplementary file 3. SLIDE scores for individual specimens. SLIDE scores (*Sen et al., 2015*) for each individual specimen were reported as the mean of the remodeling ratios corresponding to each infected cell. Scores > 1.2 revealed viral-induced remodeling at 20% or more fold change. P-values correspond to the significance of the remodeling score and are calculated as described in the Methods.

• Transparent reporting form

## Data availability

Raw CyTOF datasets have been made publicly available through the public repository Dryad as detailed in the transparent reporting form. The link for accessing these datasets is: https://doi.org/10.7272/Q6DZ06HN.

The following dataset was generated:

| Author(s) | Year | Dataset title | Dataset URL | Database and Identifier |
|---|---|---|---|---|
| Nadia Roan | 2020 | Data from: T cells from the female genital tract are highly permissive to HIV infection and remodeled by HIV to promote systemic viral spread | https://doi.org/10.7272/Q6DZ06HN | Dryad Digital Repository, 10.5061/dryad.Q6DZ06HN |

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
