## [Decision Letter]

Thank you for submitting your article "HIV efficiently infects T cells from the female genital tract and remodels them to promote systemic viral spread" for consideration by *eLife*. Your article has been reviewed by three peer reviewers, and the evaluation has been overseen by a Reviewing Editor and Wendy Garrett as the Senior Editor. The following individuals involved in review of your submission have agreed to reveal their identity: Marta Rodriguez-Garcia (Reviewer #2).

The reviewers have discussed the reviews with one another and the Reviewing Editor has drafted this decision to help you prepare a revised submission.

Summary:

In this manuscript, Ma and colleagues evaluate the susceptibility of endometrial CD4+ T cells to HIV infection. They perform in-depth phenotypic analyses of HIV-infected T cells isolated from the female reproductive tract (FRT) using a Mass Cytometry approach (CyTOF).

The authors find that endometrial T cells are more susceptible to infection than blood T cells, and that the initial targets in the endometrium are phenotypically more diverse compared to blood T cells. Additionally, they identify specific modifications induced by productive HIV-infection that may result in increased cell survival, impaired antigen recognition and viral dissemination of HIV-infected cells to lymphatic tissues. Finally, they observed that HIV infection upregulates the anti-apoptotic protein BIRC5 also in T cells from FRT as previously observed in CD4 T cells form peripheral blood (Kuo et al., 2018).

The major conclusion is, thus, that HIV reprograms the phenotypic profile of the infected cells in favor of the virus, by e. g. manipulating cell trafficking and cell survival mechanisms/cell surface markers. The study takes advantage of impressive, recently-developed technologies, including biocomputational algorithms for identifying "predicted precursors" of infected cells — a technique to infer the phenotypic profile of cells prior to infection.

In conclusion, the work is original, novel and presents three key components that differentiate this study from previous work: the use of transmitter/founder viruses, the inclusion of semen component in the infection protocol and the focus on endometrium. All three reviewers express high enthusiasm for the presented work, which represents a very notable advance to the field. The reviewers identified two very novel and unique findings that need additional work: 1) the high susceptibility of endometrial cells compared to blood cells and 2) the modification of cells following infection.

Essential revisions:

1) One of the most important findings of this manuscript is that non-stimulated endometrial CD4 T cells are much more permissive to HIV infection than activated CD4 T cells from blood. This striking finding needs be analyzed in more detail — can the authors be more specific what mechanisms are responsible for this difference? Consideration should be given to using alternative viral species (e. g. X4-tropic or VSV-G pseudotyped viruses). The authors should also show the percentage of naÐ¯ve, CM and EM in in both PBMC and T cells from endometrial biopsies. These data would give the information on how many T cells from FRT are potentially targets for infection.

2A) The key experiment missing from the study is a control demonstrating that blocking infection abrogates remodeling. The study compares uninfected vs. infected T cells, which are analyzed only at day 3 post-infection. A control including T cells exposed to HIV in the presence of antiviral drugs that prevent productive infection (blocking RT for example) would demonstrate that productive infection is necessary for the remodeling effects observed.

2B) The authors conclude that the differences between infected and uninfected cells are due to remodeling induced by productive infection. But the data is generated by pre-gating the HIV+ population and comparing it to uninfected cells, which were not exposed to the virus. How would the t-SNE analysis look like if the whole population of cells that was exposed to the virus was included in the analysis? Are the HSA negative cells also modified? This would indicate an effect of viral exposure or indirect effects induced by other cells present in the mixture instead of direct modification by productive infection.

2C) The authors should also discuss the mechanisms leading to "remodeling" or reprogramming of phenotypic properties of infected cells more clearly — is this related to cell-intrinsic innate immune recognition? Is it related to cytopathic effects? Is it related to DNA damage responses resulting from chromosomal integration?

3) It is unclear whether PBMCs and tonsil cells were incubated with SEM86 for comparisons, and whether uninfected controls were treated with SEM86. How long was SEM86 incubation? Did it modify the phenotype of the cells by itself in the absence of virus? These controls need to be shown. This is a key difference in this study compared to other studies characterizing HIV infection in the genital tract, and it should be mentioned in the Results as well.

4) The analysis done to identify the PRE cells considers the HIV-infected cells three days after infection and analyzes the uninfected control (also three days of incubation?) to find the original preferential cell infected. How is this model accounting for the potential effects of culturing the cells? The uninfected control three days after culture is probably different than the initial population of cells present when infection was performed on day 0. What if specific T cell subsets survive culture conditions better than others?

---

## [Author Response]

Essential revisions:1) One of the most important findings of this manuscript is that non-stimulated endometrial CD4 T cells are much more permissive to HIV infection than activated CD4 T cells from blood. This striking finding needs be analyzed in more detail — can the authors be more specific what mechanisms are responsible for this difference? Consideration should be given to using alternative viral species (e. g. X4-tropic or VSV-G pseudotyped viruses).

We have addressed the reviewer’s suggestion to better characterize the features associated with the high permissivity of endometrial CD4+ T cells to infection, by first comparing the expression levels of 8 different activation markers and the CCR5 co-receptor on Tm cells from the endometrium vs. unstimulated PBMCs for the four donors we had characterized in Figure 2. We found that relative to their blood counterparts, ET Tm cells expressed significantly higher levels of six of the eight activation markers (Figure 2—figure supplement 7A), and more frequently expressed the CCR5 co-receptor (Figure 2—figure supplement 7B). These results imply that both activation state and co-receptor expression may contribute to the increased permissivity of the ET Tm cells.

To assess whether the activation state of T cells alone can account for the high susceptibility of ETs to infection, we conducted an experiment where unstimulated ETs, unstimulated PBMCs, and PHA-stimulated PBMCs (as an abundant source of highly activated T cells) were exposed to F4.HSA, phenotyped by CyTOF, and analyzed by PP-SLIDE. Of note, the ETs and PBMCs in this experiment were all from the same donor. We found the following: (1) infection rates were higher in the unstimulated ETs than the PHA-stimulated PBMCs; (2) the phenotypes of the HIV-susceptible cells in unstimulated ETs were distinct from those in stimulated PBMCs; (3) multiple activation markers were expressed at higher levels on the stimulated PBMCs than the unstimulated ETs, despite the former being relatively less susceptible to infection; and (4) CCR5 levels were higher on the unstimulated ET Tm cells than stimulated PBMC Tm cells. These data are presented as the new Figure 2—figure supplement 7C-G. Together, these results suggest that T cell activation state does not fully account for the high susceptibility of ETs to infection. Instead, the high levels of CCR5 expression, and other unique aspects described in this manuscript, appear to be more important.

The authors should also show the percentage of naïve, CM and EM in in both PBMC and T cells from endometrial biopsies. These data would give the information on how many T cells from FRT are potentially targets for infection.

The requested analysis has now been conducted and is presented as the new Figure 2—figure supplement 5C. While the proportion of Tn and Tcm cells was significantly lower in the endometrium than in blood, Tem cells showed the opposite pattern. These results suggest that one mechanism rendering ET cells more susceptible than their blood counterparts is the higher frequency of Tem cells.

2A) The key experiment missing from the study is a control demonstrating that blocking infection abrogates remodeling. The study compares uninfected vs infected T cells, which are analyzed only at day 3 post-infection. A control including T cells exposed to HIV in the presence of antiviral drugs that prevent productive infection (blocking RT for example) would demonstrate that productive infection is necessary for the remodeling effects observed.

We have conducted a modified form of the suggested experiment. Instead of just adding ART to our infection assay, we first set up cultures of endometrial cells mock-treated or infected for 3 days with F4.HSA. We then took the supernatants from these two cultures and used them as a source of conditioned media, which we added to target cells isolated from the same endometrial biopsies, but in the presence of suppressive ART. The rationale for using conditioned media containing HIV, as opposed to free HIV virions, is that the conditioned media contains infection-associated factors such as cytokines and other inflammatory mediators, that may induce phenotypic changes in cells. In this experimental system, our target cells are exposed to both virions and conditioned media from infected cells, but cannot become infected with the virus due to ART. The results of these experiments, conducted on two donor specimens, revealed that endometrial Tm cells exposed to conditioned media from infected cultures in the presence of ART are not remodeled, as they exhibit lower SLIDE scores than productively-infected cells (Figure 2—figure supplement 4C), and appear in similar regions of t-SNE space as Tm cells exposed to conditioned media from uninfected cultures (Figure 2—figure supplement 4D).

2B) The authors conclude that the differences between infected and uninfected cells are due to remodeling induced by productive infection. But the data is generated by pre-gating the HIV+ population and comparing it to uninfected cells, which were not exposed to the virus. How would the t-SNE analysis look like if the whole population of cells that was exposed to the virus was included in the analysis? Are the HSA negative cells also modified? This would indicate an effect of viral exposure or indirect effects induced by other cells present in the mixture instead of direct modification by productive infection.

We have included this suggested analysis as the new Figure 2—figure supplement 4A. In contrast to the productively-infected HSA+ cells, the bystander (HSA-) Tm cells resided in regions of the t-SNE similar to regions occupied by Tm cells in the uninfected sample. Furthermore, the SLIDE scores for the HSA+ cells were significantly higher than that of the HSA- Tm cells (Figure 2—figure supplement 4B), confirming that remodeling is primarily driven by productive infection.

2C) The authors should also discuss the mechanisms leading to "remodeling" or reprogramming of phenotypic properties of infected cells more clearly — is this related to cell-intrinsic innate immune recognition? Is it related to cytopathic effects? Is it related to DNA damage responses resulting from chromosomal integration?

As described above, in contrast to productively-infected cells, bystander cells in infected cultures exhibit limited remodeling. These results suggest that the extensive remodeling that we observe requires proviral integration. The fact that bystander cells exhibit limited remodeling, and that addition of supernatants from infected cultures whilst blocking productive infection with antiretroviral drugs does not induce remodeling in the Tm cells (see above), suggests that the inflammatory cytokine milieu is not sufficient to remodel cells. Rather, it is likely that cell-intrinsic innate immune recognition and/or DNA damage responses, as well as viral-induced remodeling through the activity of viral accessory genes, drives the remodeling. Cytopathic effects are unlikely contributing as we had gated on live cells in our analyses. These concepts are now brought forth in the Discussion (paragraph three).

3) It is unclear whether PBMCs and tonsil cells were incubated with SEM86 for comparisons, and whether uninfected controls were treated with SEM86.

In the study comparing PBMCs and tonsil cells'responses to SEM86, all samples (including the uninfected ones) were treated with SEM86. This has now been clarified in the Results section (paragraph one).

How long was SEM86 incubation?

We apologize for the lack of details on the experimental protocol implemented for SEM86 treatment. We had incubated the virus for 15-30 minutes with 100 μg/ml SEM86, followed by dilution of this viral stock 10-fold onto target cells. The details of the protocol have now been added to the Materials and methods section (subsection "Virus Preparation and infection assays").

Did it modify the phenotype of the cells by itself in the absence of virus?

We believe that SEM86 did not remodel the cells independent of virus, because limited remodeling was observed in the bystander cells in the culture that were exposed to the same amount of SEM86 (see above). Consistent with this notion, in our FACS experiments the mean fluorescence intensity (MFI) of CD3 and CD4 was not significantly different between T cells treated or not with SEM86. These data are shown in Author response image 1.

**Author response image 1. respfig1:** SEM86 does not alter the surface expression levels of CD3 or CD4 in primary T cells.

These controls need to be shown. This is a key difference in this study compared to other studies characterizing HIV infection in the genital tract, and it should be mentioned in the Results as well.

The addition of SEM86 to our cultures has now been mentioned in the Results section (paragraph one).

4) The analysis done to identify the PRE cells considers the HIV-infected cells three days after infection and analyzes the uninfected control (also three days of incubation?) to find the original preferential cell infected. How is this model accounting for the potential effects of culturing the cells? The uninfected control three days after culture is probably different than the initial population of cells present when infection was performed on day 0. What if specific T cell subsets survive culture conditions better than others?

The uninfected control was indeed mock-treated for 3 days. This has now been clarified in the Materials and methods (subsection "Virus Preparation and infection assays"). We opted to not compare the infected cells from day 3 to uninfected cells from day 0, since if that were done the phenotypic changes in the infected cells would reflect not only changes induced by infection, but also phenotypic changes that incur during ex vivo culture of primary cells, a process known as phenotypic drift. Such changes could include, as the reviewer mentions, the preferential survival of particular T cell subsets. To confirm that drift did occur in ETs, we conducted CyTOF analysis on an endometrial specimen immediately after generating the single-cell suspension of the freshly isolated specimen (day 0), and after 3 days of culturing the specimen in media (day 3). A t-SNE analysis of the resulting CyTOF datasets demonstrated differences between Tm cells of the two specimens, as revealed by the cells occupying unique regions of the t-SNE space, see Author response image 2.

**Author response image 2. respfig2:** Phenotypic drift occurs upon culturing of freshly isolated endometrial T cells for 3 days.

These results suggest that drift does indeed occur in ETs, but is an unavoidable issue with any kind of ex vivo viral infection experiments using primary cells targets. Of note, the phenotypic differences between the Day 0 and Day 3 specimens may also be in part due to the lack of time for the Day 0 specimen to re-express antigens following collagenase digestion. To recognize drift as an unavoidable limitation of our experimental model, we have now brought up the issue in the Discussion, as well as how we tried to account for it by always comparing infected samples to uninfected ones cultured for the same period of time (paragraph eight).